# PROVABLY FASTER ALGORITHMS FOR BILEVEL OPTIMIZATION AND APPLICATIONS TO META-LEARNING

## ABSTRACT

Bilevel optimization has arisen as a powerful tool for many machine learning problems such as meta-learning, hyperparameter optimization, and reinforcement learning. In this paper, we investigate the nonconvex-strongly-convex bilevel optimization problem. For deterministic bilevel optimization, we provide a comprehensive finite-time convergence analysis for two popular algorithms respectively based on approximate implicit differentiation (AID) and iterative differentiation (ITD). For the AID-based method, we orderwisely improve the previous finite-time convergence analysis due to a more practical parameter selection as well as a warm start strategy, and for the ITD-based method we establish the first theoretical convergence rate. Our analysis also provides a quantitative comparison between ITD and AID based approaches. For stochastic bilevel optimization, we propose a novel algorithm named stocBiO, which features a sample-efficient hypergradient estimator using efficient Jacobian- and Hessian-vector product computations. We provide the finite-time convergence guarantee for stocBiO, and show that stocBiO outperforms the best known computational complexities orderwisely with respect to the condition number $\kappa$ and the target accuracy $\epsilon$. We further validate our theoretical results and demonstrate the efficiency of bilevel optimization algorithms by the experiments on meta-learning and hyperparameter optimization.

## 1 INTRODUCTION

Bilevel optimization has received significant attention recently and become an influential framework in various machine learning applications including meta-learning (Franceschi et al., 2018; Bertinetto et al., 2018; Rajeswaran et al., 2019; Ji et al., 2020a), hyperparameter optimization (Franceschi et al., 2018; Shaban et al., 2019; Feurer & Hutter, 2019), reinforcement learning (Konda & Tsitsiklis, 2000; Hong et al., 2020), and signal processing (Kunapuli et al., 2008; Flamary et al., 2014). A general bilevel optimization takes the following formulation.

$$\min_{x \in \mathbb{R}^p} \Phi(x) := f(x, y^*(x)) \quad \text{s.t.} \quad y^*(x) = \arg\min_{y \in \mathbb{R}^q} g(x, y), \tag{1}$$

where the upper- and inner-level functions $f$ and $g$ are both jointly continuously differentiable. The goal of eq. (1) is to minimize the objective function $\Phi(x)$ w.r.t. $x$, where $y^*(x)$ is obtained by solving the lower-level minimization problem. In this paper, we focus on the setting where the lower-level function $g$ is strongly convex with respect to (w.r.t.) $y$, and the upper-level objective function $\Phi(x)$ is nonconvex w.r.t. $x$. Such types of geometrics commonly exist in many applications including meta-learning and hyperparameter optimization, where $g$ corresponds to an empirical loss with a strongly-convex regularizer and $x$ are parameters of neural networks.

A broad collection of algorithms have been proposed to solve such types of bilevel optimization problems. For example, Hansen et al. (1992); Shi et al. (2005); Moore (2010) reformulated the bilevel problem in eq. (1) into a single-level constrained problem based on the optimality conditions of the lower-level problem. However, such type of methods often involve a large number of constraints, and are hard to implement in machine learning applications. Recently, more efficient gradient-based bilevel optimization algorithms have been proposed, which can be generally categorized into the approximate implicit differentiation (AID) based approach (Domke, 2012; Pedregosa, 2016; Gould et al., 2016; Liao et al., 2018; Ghadimi & Wang, 2018; Grazzi et al., 2020; Lorraine

et al., 2020) and the iterative differentiation (ITD) based approach (Domke, 2012; Maclaurin et al., 2015; Franceschi et al., 2017; 2018; Shaban et al., 2019; Grazzi et al., 2020). However, most of these studies have focused on the asymptotic convergence analysis, and the finite-time analysis (that characterizes how fast an algorithm converges) has not been well explored except a few attempts recently. Ghadimi & Wang (2018) provided the finite-time analysis for the ITD-based approach. Grazzi et al. (2020) provided the iteration complexity for the hypergradient computation via ITD and AID, but did not characterize the finite-time convergence for the entire execution of algorithms.

- Thus, the first focus of this paper is to develop a *comprehensive and enhanced* theory, which covers a broader class of bilevel optimizers via ITD and AID based techniques, and more importantly, to improve the exiting analysis with a more practical parameter selection and order-level *lower* computational complexity.

The *stochastic* bilevel optimization often occurs in applications where fresh data need to be sampled as the algorithms run (e.g., reinforcement learning (Hong et al., 2020)) or the sample size of training data is large (e.g., hyperparameter optimization (Franceschi et al., 2018), Stackelberg game (Roth et al., 2016)). Typically, the corresponding objective function is given by

$$\min_{x \in \mathbb{R}^p} \Phi(x) = f(x, y^*(x)) := \begin{cases} \mathbb{E}_\xi \left[ F(x, y^*(x); \xi) \right] \\ \frac{1}{n} \sum_{i=1}^n F(x, y^*(x); \xi_i) \end{cases}$$

$$\text{s.t.} \quad y^*(x) = \arg\min_{y \in \mathbb{R}^q} g(x, y) := \begin{cases} \mathbb{E}_\zeta \left[ G(x, y^*(x); \zeta) \right] \\ \frac{1}{m} \sum_{i=1}^m G(x, y^*(x); \zeta_i), \end{cases} \tag{2}$$

where $f(x, y)$ and $g(x, y)$ take either the expectation form w.r.t. the random variables $\xi$ and $\zeta$ or the finite-sum form over given data $\mathcal{D}_{n,m} = \{\xi_i, \zeta_j, i = 1, ..., n; j = 1, ..., m\}$ often with large sizes $n$ and $m$. During the optimization process, the algorithms sample data batch via the distributions of $\xi$ and $\zeta$ or from the set $\mathcal{D}_{n,m}$. For such a stochastic setting, Ghadimi & Wang (2018) proposed a bilevel stochastic approximation (BSA) method via single-sample gradient and Hessian estimates. Based on such a method, Hong et al. (2020) further proposed a two-timescale stochastic approximation (TTSA), and showed that TTSA achieves a better trade-off between the complexities of inner- and outer-loop optimization stages than BSA.

- The second focus of this paper is to design a more sample-efficient algorithm for bilevel stochastic optimization, which achieves an order-level lower computational complexity over BSA and TTSA.

## 1.1 MAIN CONTRIBUTIONS

Our main contributions lie in developing enhanced theory and provably faster algorithms for the nonconvex-strongly-convex bilevel deterministic and stochastic optimization problems, respectively. Our analysis involves several new developments, which can be of independent interest.

We first provide a unified finite-time convergence and complexity analysis for both ITD and AID based bilevel optimizers, which we call as ITD-BiO and AID-BiO. Compared to existing analysis in Ghadimi & Wang (2018) for AID-BiO that requires a continuously increasing number of inner-loop steps to achieve the guarantee, our analysis allows a constant number of inner-loop steps as often used in practice. In addition, we introduce a warm start initialization for the inner-loop updates and the outer-loop hypergradient estimation, which allows us to backpropagate the tracking errors to previous loops, and results in an improved computational complexity. As shown in Table 1, the gradient complexities Gc($f, \epsilon$), Gc($g, \epsilon$), and Jacobian- and Hessian-vector product complexities JV($g, \epsilon$) and HV($g, \epsilon$) of AID-BiO to attain an $\epsilon$-accurate stationary point improve those of Ghadimi & Wang (2018) by the order of $\kappa$, $\kappa\epsilon^{-1/4}$, $\kappa$, and $\kappa$, respectively, where $\kappa$ is the condition number. In addition, our analysis shows that AID-BiO requires less computations of Jacobian- and Hessian-vector products than ITD-BiO by an order of $\kappa$ and $\kappa^{1/2}$, which provides a justification for the observation in Grazzi et al. (2020) that ITD often has a larger memory cost than AID.

We then propose a stochastic bilevel optimizer (stocBiO) to solve the stochastic bilevel optimization problem in eq. (2). Our algorithm features a *mini-batch* hyper-gradient estimation via implicit differentiation, where the core design involves a sample-efficient Hypergradient estimator via the Neumann series. As shown in Table 2, the gradient complexities of our proposed algorithm w.r.t. $F$

Table 1: Comparison of bilevel deterministic optimization algorithms.

| Algorithm | $\text{Gc}(f, \epsilon)$ | $\text{Gc}(g, \epsilon)$ | $\text{JV}(g, \epsilon)$ | $\text{HV}(g, \epsilon)$ |
|---|---|---|---|---|
| AID-BiO (Ghadimi & Wang, 2018) | $\mathcal{O}(\kappa^4\epsilon^{-1})$ | $\mathcal{O}(\kappa^5\epsilon^{-5/4})$ | $\mathcal{O}\left(\kappa^4\epsilon^{-1}\right)$ | $\widetilde{\mathcal{O}}\left(\kappa^{4.5}\epsilon^{-1}\right)$ |
| AID-BiO (this paper) | $\mathcal{O}(\kappa^3\epsilon^{-1})$ | $\mathcal{O}(\kappa^4\epsilon^{-1})$ | $\mathcal{O}\left(\kappa^3\epsilon^{-1}\right)$ | $\mathcal{O}\left(\kappa^{3.5}\epsilon^{-1}\right)$ |
| ITD-BiO (this paper) | $\mathcal{O}(\kappa^3\epsilon^{-1})$ | $\widetilde{\mathcal{O}}(\kappa^4\epsilon^{-1})$ | $\widetilde{\mathcal{O}}\left(\kappa^4\epsilon^{-1}\right)$ | $\widetilde{\mathcal{O}}\left(\kappa^4\epsilon^{-1}\right)$ |

$\text{Gc}(f, \epsilon)$ and $\text{Gc}(g, \epsilon)$: number of gradient evaluations w.r.t. $f$ and $g$. $\kappa$ : the condition number.
$\text{JV}(g, \epsilon)$: number of Jacobian-vector products $\nabla_x \nabla_y g(x, y)v$. Notation $\widetilde{\mathcal{O}}$: omit $\log \frac{1}{\epsilon}$ terms.
$\text{HV}(g, \epsilon)$: number of Hessian-vector products $\nabla_y^2 g(x, y)v$.

Table 2: Comparison of bilevel stochastic optimization algorithms.

| Algorithm | $\text{Gc}(F, \epsilon)$ | $\text{Gc}(G, \epsilon)$ | $\text{JV}(G, \epsilon)$ | $\text{HV}(G, \epsilon)$ |
|---|---|---|---|---|
| TTSA (Hong et al., 2020) | $\mathcal{O}(\text{poly}(\kappa)\epsilon^{-\frac{5}{2}})^*$ | $\mathcal{O}(\text{poly}(\kappa)\epsilon^{-\frac{5}{2}})$ | $\mathcal{O}(\text{poly}(\kappa)\epsilon^{-\frac{5}{2}})$ | $\mathcal{O}(\text{poly}(\kappa)\epsilon^{-\frac{5}{2}})$ |
| BSA (Ghadimi & Wang, 2018) | $\mathcal{O}(\kappa^6\epsilon^{-2})$ | $\mathcal{O}(\kappa^9\epsilon^{-3})$ | $\mathcal{O}\left(\kappa^6\epsilon^{-2}\right)$ | $\widetilde{\mathcal{O}}\left(\kappa^6\epsilon^{-2}\right)$ |
| stocBiO (this paper) | $\mathcal{O}(\kappa^5\epsilon^{-2})$ | $\mathcal{O}(\kappa^9\epsilon^{-2})$ | $\mathcal{O}\left(\kappa^5\epsilon^{-2}\right)$ | $\widetilde{\mathcal{O}}\left(\kappa^6\epsilon^{-2}\right)$ |

$^*$ We use $\text{poly}(\kappa)$ because Hong et al. (2020) does not provide the explicit dependence on $\kappa$.

and $G$ improve upon those of BSA (Ghadimi & Wang, 2018) by an order of $\kappa$ and $\epsilon^{-1}$, respectively. In addition, the Jacobian-vector product complexity $\text{JV}(G, \epsilon)$ of our algorithm improves that of BSA by an order of $\kappa$. In terms of the target accuracy $\epsilon$, our computational complexities improve those of TTSA (Hong et al., 2020) by an order of $\epsilon^{-1/2}$.

We further provide the theoretical complexity guarantee of ITD-BiO, AID-BiO and stocBiO in meta-learning and hyperparameter optimization. The experiments validate our theoretical results for determinisitic bilevel optimization, and demonstrate the superior efficiency of stocBiO for stochastic bilevel optimization. Due to the space limitations, we present all theoretical and empirical results on hyperparameter optimization in the supplementary materials.

## 1.2 RELATED WORK

**Bilevel optimization approaches**: Bilevel optimization was first introduced by Bracken & McGill (1973). Since then, a number of bilevel optimization algorithms have been proposed, which include but not limited to constraint-based methods (Shi et al., 2005; Moore, 2010) and gradient-based methods (Domke, 2012; Pedregosa, 2016; Gould et al., 2016; Maclaurin et al., 2015; Franceschi et al., 2018; Ghadimi & Wang, 2018; Liao et al., 2018; Shaban et al., 2019; Hong et al., 2020; Liu et al., 2020; Li et al., 2020; Grazzi et al., 2020; Lorraine et al., 2020). Among them, Ghadimi & Wang (2018); Hong et al. (2020) provided the finite-time complexity analysis for their proposed methods for the nonconvex-strongly-convex bilevel optimization problem. For such a problem, this paper develops a general and enhanced finite-time analysis for gradient-based bilevel optimizers for the deterministic setting, and proposes a novel algorithm for the stochastic setting with order-level lower computational complexity than the existing results.

Some works have studied other types of loss geometries. For example, Liu et al. (2020); Li et al. (2020) assumed that the lower- and upper-level functions $g(x, \cdot)$ and $f(x, \cdot)$ are convex and strongly-convex, and provided an asymptotic analysis for their methods. Ghadimi & Wang (2018); Hong et al. (2020) studied the setting where $\Phi(\cdot)$ is strongly-convex or convex, and $g(x, \cdot)$ is strongly-convex.

**Bilevel optimization in meta-learning**: Bilevel optimization framework has been successfully employed in meta-learning recently (Snell et al., 2017; Franceschi et al., 2018; Rajeswaran et al., 2019; Zügner & Günnemann, 2019; Ji et al., 2020a;b). For example, Snell et al. (2017) proposed a bilevel optimization procedure for meta-learning to learn a common embedding model for all tasks. Rajeswaran et al. (2019) reformulated the model-agnostic meta-learning (MAML) (Finn et al., 2017) as a bilevel optimization problem, and proposed iMAML via implicit gradient. The paper provides a theoretical guarantee for two popular types of bilevel optimization algorithms, i.e., AID-BiO and ITD-BiO, for meta-learning.

**Bilevel optimization in hyperparameter optimization**: Hyperparameter optimization has become increasingly important as a powerful tool in the automatic machine learning (autoML) (Okuno et al.,

---

**Algorithm 1** Deterministic bilevel optimization via AID or ITD

---

1: **Input:** Stepsizes $\alpha, \beta > 0$, initializations $x_0, y_0, v_0$.
2: **for** $k = 0, 1, 2, ..., K$ **do**
3:     Set $y_k^0 = y_{k-1}^T$ if $k > 0$ and $y_0$ otherwise
4:     **for** $t = 1, ...., T$ **do**
5:         Update $y_k^t = y_k^{t-1} - \alpha \nabla_y g(x_k, y_k^{t-1})$
6:     **end for**
7:     Hypergradient estimation via
      •   AID: 1) set $v_k^0 = v_{k-1}^N$ if $k > 0$ and $v_0$ otherwise
              2) solve $v_k^N$ from $\nabla_y^2 g(x_k, y_k^T)v = \nabla_y f(x_k, y_k^T)$ via $N$ steps of CG starting from $v_k^0$
              3) compute Jacobian-vector product $\nabla_x \nabla_y g(x_k, y_k^T)v_k^N$ via automatic differentiation
              4) compute $\widehat{\nabla}\Phi(x_k) = \nabla_x f(x_k, y_k^T) - \nabla_x \nabla_y g(x_k, y_k^T)v_k^N$
      •   ITD: compute $\widehat{\nabla}\Phi(x_k) = \frac{\partial f(x_k, y_k^T)}{x_k}$ via backpropagation w.r.t. $x_k$
8:     Update $x_{k+1} = x_k - \beta \frac{\partial f(x_k, y_k^T)}{\partial x_k}$
9: **end for**

---

2018; Yu & Zhu, 2020). Recently, various bilevel optimization algorithms have been proposed in the context of hyperparameter optimization, which include implicit differentiation based methods (Pedregosa, 2016), dynamical system based methods via reverse or forward gradient computation (Franceschi et al., 2017; 2018; Shaban et al., 2019), etc. This paper demonstrates the superior efficiency of the proposed stocBiO algorithm in hyperparameter optimization.

## 2   Algorithms

In this section, we describe two popular types of *deterministic* bilevel optimization algorithms, and propose a new algorithms for *stochastic* bilevel optimization.

### 2.1   Algorithms for Deterministic Bilevel Optimization

As shown in Algorithm 1, we describe two popular types of deterministic bilevel optimizers respectively based on AID and ITD (referred to as AID-BiO and ITD-BiO) for solving the problem eq. (1).

Both AID-BiO and ITD-BiO update in a nested-loop manner. In the inner loop, both of them run $T$ steps of gradient decent (GD) to find an approximation point $y_k^T$ close to $y^*(x_k)$. Note that we choose the initialization $y_k^0$ of each inner loop as the output $y_{k-1}^T$ of the preceding inner loop rather than a random start. Such a *warm start* allows us to backpropagate the tracking error $\|y_k^T - y^*(x_k)\|$ to previous loops, and yields an improved computational complexity.

At the outer loop, AID-BiO first solves $v_k^N$ from a linear system $\nabla_y^2 g(x_k, y_k^T)v = \nabla_y f(x_k, y_k^T)$[1] using $N$ steps of conjugate-gradient (CG) starting from $v_k^0$ (where we also adopt a warm start scheme here by setting $v_k^0 = v_{k-1}^N$), and then constructs

$$\widehat{\nabla}\Phi(x_k) = \nabla_x f(x_k, y_k^T) - \nabla_x \nabla_y g(x_k, y_k^T)v_k^N \tag{3}$$

as an estimate of the true hypergradient $\nabla\Phi(x_k)$, whose form is given by the following proposition.

**Proposition 1.** *Recalling the definition* $\Phi(x) := f(x, y^*(x))$, *it holds that*

$$\nabla\Phi(x_k) = \nabla_x f(x_k, y^*(x_k)) - \nabla_x \nabla_y g(x_k, y^*(x_k))v_k^*, \tag{4}$$

*where $v_k^*$ is the solution of the linear system* $\nabla_y^2 g(x_k, y^*(x_k))v = \nabla_y f(x_k, y^*(x_k))$.

As shown in Domke (2012); Grazzi et al. (2020), the construction of eq. (3) involves only Hessian-vector products in solving $v_N$ via CG and Jacobian-vector product $\nabla_x \nabla_y g(x_k, y_k^T)v_k^N$, which can be efficiently computed and stored via existing automatic differentiation packages.

As a comparison, the outer loop of ITD-BiO computes the gradient $\frac{\partial f(x_k, y_k^T(x_k))}{\partial x_k}$ as an approximation of the hyper-gradient $\nabla\Phi(x_k) = \frac{\partial f(x_k, y^*(x_k))}{\partial x_k}$ via backpropagation, where we write $y_k^T(x_k)$

---

[1]This is equivalent to solve a quadratic programing $\min_v \frac{1}{2}v^T \nabla_y^2 g(x_k, y_k^T)v - v^T \nabla_y f(x_k, y_k^T)$.

---

**Algorithm 2** Stochastic bilevel optimizer (stocBiO)

---

1: **Input:** Inner- and outer-loop stepsizes $\alpha, \beta > 0$, initializations $x_0$ and $y_0$.
2: **for** $k = 0, 1, 2, ..., K$ **do**
3:     Set $y_k^0 = y_{k-1}^T$ if $k > 0$ and $y_0$ otherwise
4:     **for** $t = 1, ...., T$ **do**
5:         Draw a sample batch $\mathcal{S}_{t-1}$
6:         Update $y_k^t = y_k^{t-1} - \alpha \nabla_y G(x_k, y_k^{t-1}; \mathcal{S}_{t-1})$
7:     **end for**
8:     Draw sample batch $\mathcal{D}_F$, and compute $v_0 = \nabla_y F(x_k, y_k^T; \mathcal{D}_F)$
9:     Draw sample batch $\mathcal{D}_H$, and construct $v_Q$ via Algorithm 3
10:    Draw sample batch $\mathcal{D}_G$, and compute Jacobian-vector product $\nabla_x \nabla_y G(x_k, y_k^T; \mathcal{D}_G) v_Q$
11:    Compute gradient estimate $\widehat{\nabla} \Phi(x_k)$ via eq. (6)
12:    Update $x_{k+1} = x_k - \beta \widehat{\nabla} \Phi(x_k)$
13: **end for**

---

**Algorithm 3** Construct $v_Q$ given $v_0$

---

1: **Input:** An integer $Q$, data samples $\mathcal{D}_H = \{\mathcal{B}_j\}_{j=1}^Q$ and a constant $\eta > 0$.
2: **for** $j = 1, 2, ..., Q$ **do**
3:     Sample $\mathcal{B}_j$ and compute gradient $G_j(y) = y - \eta \nabla_y G(x, y; \mathcal{B}_j)$
4: **end for**
5: Set $r_Q = v_0$
6: **for** $i = Q, ..., 1$ **do**
7:     $r_{i-1} = \partial\big(G_i(y) r_i\big)/\partial y = r_i - \eta \nabla_y^2 G(x, y; \mathcal{B}_i) r_i$ via automatic differentiation
8: **end for**
9: Return $v_Q = \eta \sum_{i=0}^Q r_i$

---

because the output $y_k^T$ of the inner loop has a dependence on $x_k$ through the inner-loop iterative GD updates. The explicit form of the estimate $\frac{\partial f(x_k, y_k^T(x_k))}{\partial x_k}$ is given by the following proposition via the chain rule. For notation simplification, let $\prod_{j=T}^{T-1}(\cdot) = I$.

**Proposition 2.** *The gradient $\frac{\partial f(x_k, y_k^T(x_k))}{\partial x_k}$ takes the following analytical form:*

$$\frac{\partial f(x_k, y_k^T)}{\partial x_k} = \nabla_x f(x_k, y_k^T) - \alpha \sum_{t=0}^{T-1} \nabla_x \nabla_y g(x_k, y_k^t) \prod_{j=t+1}^{T-1} (I - \alpha \nabla_y^2 g(x_k, y_k^j)) \nabla_y f(x_k, y_k^T).$$

Proposition 2 shows that the differentiation involves the computations of second-order derivatives such as Hessian $\nabla_y^2 g(\cdot, \cdot)$. Since efficient Hessian-free methods such as CG have been successfully deployed in the existing automatic differentiation tools, computing these second-order derivatives reduces to more efficient computations of Jacobian- and Hessian-vector products.

## 2.2 ALGORITHM FOR STOCHASTIC BILEVEL OPTIMIZATION

We propose a new stochastic bilevel optimizer (stocBiO) in Algorithm 2 to solve the problem eq. (2). It has a double-loop structure similar to Algorithm 1, but runs $T$ steps of stochastic gradient decent (SGD) at the inner loop to obtain an approximated solution $y_k^T$. Based on the output $y_k^T$ of the inner loop, stocBiO first computes a gradient $\nabla_y F(x_k, y_k^T; \mathcal{D}_F)$ over a sample batch $\mathcal{D}_F$, and then computes a vector $v_Q$ via Algorithm 3, which takes a form of

$$v_Q = \eta \sum_{q=-1}^{Q-1} \prod_{j=Q-q}^{Q} (I - \eta \nabla_y^2 G(x_k, y_k^T; \mathcal{B}_j)) \nabla_y F(x_k, y_k^T; \mathcal{D}_F), \tag{5}$$

where $\{\mathcal{B}_j, j = 1, ..., Q\}$ are mutually-independent sample sets, $Q$ and $\eta$ are constants, and we let $\prod_{Q+1}^Q (\cdot) = I$ for notational simplification. Note that our construction of $v_Q$, i.e., Algorithm 3, is motived by the Neumann series $\sum_{i=0}^\infty U^k = (I - U)^{-1}$, and involves only Hessian-vector products rather than Hessians, and hence is computationally and memory efficient. Then, we construct

$$\widehat{\nabla} \Phi(x_k) = \nabla_x F(x_k, y_k^T; \mathcal{D}_F) - \nabla_x \nabla_y G(x_k, y_k^T; \mathcal{D}_G) v_Q \tag{6}$$

as an estimate of hypergradient $\nabla \Phi(x_k)$ given by Proposition 1. An important component of our algorithm is $v_Q$, which serves as an estimate of $v_k^*$ in eq. (4) . Compared to the deterministic case, designing a sample-efficient Hypergradient estimator in the stochastic case is more challenging. For example, instead of choosing the same batch sizes for all $\mathcal{B}_j, j = 1, ..., Q$ in eq. (5), our analysis captures the different impact of components $\nabla_y^2 G(x_k, y_k^T; \mathcal{B}_j), j = 1, ..., Q$ on the Hypergradient estimation variance, and inspires an adaptive and more efficient choice by setting $|\mathcal{B}_{Q-j}|$ to *decay exponentially* with $j$ from $0$ to $Q - 1$. By doing so, we achieve an improved complexity.

## 3 DEFINITIONS AND ASSUMPTIONS

Let $z = (x, y)$ denote all parameters. For simplicity, suppose sample sets $\mathcal{S}_t$ for all $t = 0, ..., T - 1$, $\mathcal{D}_G$ and $\mathcal{D}_F$ have the sizes of $S$, $D_g$ and $D_f$, respectively. In this paper, we focus on the following types of loss functions for both the deterministic and stochastic cases.

**Assumption 1.** *The lower-level function $g(x, y)$ is $\mu$-strongly-convex w.r.t. $y$ and the total objective function $\Phi(x) = f(x, y^*(x))$ is nonconvex w.r.t. $x$. For the stochastic setting, the same assumptions hold for $G(x, y; \zeta)$ and $\Phi(x)$, respectively.*

Since the objective function $\Phi(x)$ is nonconvex, algorithms are expected to find an $\epsilon$-accurate stationary point defined as follows.

**Definition 1.** *We say $\bar{x}$ is an $\epsilon$-accurate stationary point for the objective function $\Phi(x)$ in eq. (2) if $\mathbb{E}\|\nabla \Phi(\bar{x})\|^2 \leq \epsilon$, where $\bar{x}$ is the output of an algorithm.*

In order to compare the performance of different bilevel algorithms, we adopt the following metrics of computational complexity.

**Definition 2.** *For a function $f(x, y)$ and a vector $v$, let $\mathrm{Gc}(f, \epsilon)$ be the number of the partial gradient $\nabla_x f$ or $\nabla_y f$, and let $\mathrm{JV}(g, \epsilon)$ and $\mathrm{HV}(g, \epsilon)$ be the number of Jacobian-vector products $\nabla_x \nabla_y g(x, y)v$. and Hessian-vector products $\nabla_y^2 g(x, y)v$. For the stochastic case, similar metrics are adopted but w.r.t. the stochastic function $F(x, y; \xi)$.*

We take the following standard assumptions on the loss functions in eq. (2), which have been widely adopted in bilevel optimization (Ghadimi & Wang, 2018; Ji et al., 2020a).

**Assumption 2.** *The loss function $f(z)$ and $g(z)$ satisfy*

- *$f(z)$ is $M$-Lipschitz, i.e., for any $z, z'$, $|f(z) - f(z')| \leq M\|z - z'\|$.*

- *Gradients $\nabla f(z)$ and $\nabla f(z)$ are $L$-Lipschitz, i.e., for any $z, z'$,*
$$\|\nabla f(z) - \nabla f(z')\| \leq L\|z - z'\|, \ \|\nabla g(z) - \nabla g(z')\| \leq L\|z - z'\|.$$

*For the stochastic case, the same assumptions hold for $F(z; \xi)$ and $G(z; \zeta)$ for any given $\xi$ and $\zeta$.*

As shown in Proposition 1, the gradient of the objective function $\Phi(x)$ involves the second-order derivatives $\nabla_x \nabla_y g(z)$ and $\nabla_y^2 g(z)$. The following assumption imposes the Lipschitz conditions on such high-order derivatives, as also made in Ghadimi & Wang (2018).

**Assumption 3.** *Suppose the derivatives $\nabla_x \nabla_y g(z)$ and $\nabla_y^2 g(z)$ are $\tau$- and $\rho$- Lipschitz, i.e.,*

- *For any $z, z'$, $\|\nabla_x \nabla_y g(z) - \nabla_x \nabla_y g(z')\| \leq \tau\|z - z'\|$.*

- *For any $z, z'$, $\|\nabla_y^2 g(z) - \nabla_y^2 g(z')\| \leq \rho\|z - z'\|$.*

*For the stochastic case, the same assumptions hold for $\nabla_x \nabla_y G(z; \zeta)$ and $\nabla_y^2 G(z; \zeta)$ for any $\zeta$.*

As typically adopted in the analysis for stochastic optimization, we make the following bounded-variance assumption for the lower-level stochastic function $G(z; \zeta)$.

**Assumption 4.** *$\nabla G(z; \zeta)$ has a bounded variance, i.e., $\mathbb{E}_\xi \|\nabla G(z; \zeta) - \nabla g(z)\|^2 \leq \sigma^2$ for some $\sigma$.*

## 4 MAIN RESULTS FOR BILEVEL OPTIMIZATION

### 4.1 DETERMINISTIC BILEVEL OPTIMIZATION

We first characterize the convergence and complexity performance of the AID-BiO algorithm. Let $\kappa = \frac{L}{\mu}$ denote the condition number.

**Theorem 1** (AID-BiO). *Suppose Assumptions 1, 2, 3 hold. Define a smoothness parameter $L_\Phi = L + \frac{2L^2 + \tau M^2}{\mu} + \frac{\rho LM + L^3 + \tau ML}{\mu^2} + \frac{\rho L^2 M}{\mu^3} = \Theta(\kappa^3)$, choose the stepsizes $\alpha \leq \frac{1}{L}$, $\beta = \frac{1}{8L_\Phi}$, and set the inner-loop iteration number $T \geq \Theta(\kappa)$ and the CG iteration number $N \geq \Theta(\sqrt{\kappa})$, where the detailed forms of $T, N$ can be found in Appendix E. Then, the outputs of AID-BiO satisfy*

$$\frac{1}{K} \sum_{k=0}^{K-1} \|\nabla \Phi(x_k)\|^2 \leq \frac{64 L_\Phi (\Phi(x_0) - \inf_x \Phi(x)) + 5\Delta_0}{K}, \tag{7}$$

*where $\Delta_0 = \|y_0 - y^*(x_0)\|^2 + \|v_0^* - v_0\|^2 > 0$.*

*In order to achieve an $\epsilon$-accurate stationary point, we have*

- *Gradient complexity:* $\mathrm{Gc}(f, \epsilon) = \mathcal{O}(\kappa^3 \epsilon^{-1}), \mathrm{Gc}(g, \epsilon) = \mathcal{O}(\kappa^4 \epsilon^{-1})$.

- *Jacobian- and Hessian-vector product:* $\mathrm{JV}(g, \epsilon) = \mathcal{O}\left(\kappa^3 \epsilon^{-1}\right), \mathrm{HV}(g, \epsilon) = \mathcal{O}\left(\kappa^{3.5} \epsilon^{-1}\right)$.

It can be seen from Table 1 that the complexities $\mathrm{Gc}(f, \epsilon), \mathrm{Gc}(g, \epsilon), \mathrm{JV}(g, \epsilon)$ and $\mathrm{HV}(g, \epsilon)$ of our analysis improves that of Ghadimi & Wang (2018) (eq. (2.30) therein) by the order of $\kappa$, $\kappa\epsilon^{-1/4}$, $\kappa$ and $\kappa$. Such an improvement is achieved by a refined analysis with a constant number of inner-loop steps, and by a warm start strategy to backpropagate the tracking errors $\|y_k^T - y^*(x_k)\|$ and $\|v_k^N - v_k^*\|$ to previous loops, as also demonstrated by our meta-learning experiments.

We next characterize the convergence and complexity performance of the ITD-BiO algorithm.

**Theorem 2** (ITD-BiO). *Suppose Assumptions 1, 2, and 3 hold. Define the parameter $L_\Phi$ as in Theorem 1, and choose $\alpha \leq \frac{1}{L}$, $\beta = \frac{1}{4L_\Phi}$ and $T \geq \Theta(\kappa \log \frac{1}{\epsilon})$, where the detailed form of $T$ can be found in Appendix F. Then, the outputs of ITD-BiO satisfy*

$$\frac{1}{K} \sum_{k=0}^{K-1} \|\nabla \Phi(x_k)\|^2 \leq \frac{16 L_\Phi (\Phi(x_0) - \inf_x \Phi(x))}{K} + \frac{2\epsilon}{3}.$$

*In order to achieve an $\epsilon$-accurate stationary point, we have*

- *Gradient complexity:* $\mathrm{Gc}(f, \epsilon) = \mathcal{O}(\kappa^3 \epsilon^{-1}), \mathrm{Gc}(g, \epsilon) = \mathcal{O}(\kappa^4 \epsilon^{-1} \log(\frac{1}{\epsilon}))$.

- *Jacobian- and Hessian-vector product complexity:*

$$\mathrm{JV}(g, \epsilon) = \mathcal{O}\left(\kappa^4 \epsilon^{-1} \log \epsilon^{-1}\right), \mathrm{HV}(g, \epsilon) = \mathcal{O}\left(\kappa^4 \epsilon^{-1} \log \epsilon^{-1}\right).$$

By comparing Theorem 1 and Theorem 2, it can be seen that the complexities $\mathrm{Gc}(g, \epsilon), \mathrm{JV}(g, \epsilon)$, and $\mathrm{HV}(g, \epsilon)$ of AID-BiO are better than those of ITD-BiO by the order of $\log(\frac{1}{\epsilon})$, $\kappa \log(\frac{1}{\epsilon})$ and $\kappa^{0.5} \log(\frac{1}{\epsilon})$. This is in consistence with the comparison in Grazzi et al. (2020) that AID-BiO often has a lower memory cost than ITD-BiO.

## 4.2 STOCHASTIC BILEVEL OPTIMIZATION

We first characterize the bias and variance of an important component $v_Q$ in eq. (5).

**Proposition 3.** *Suppose Assumptions 1, 2 and 3 hold. Let the constant $\eta \leq \frac{1}{L}$ and choose the batch sizes $|\mathcal{B}_{Q+1-j}| = BQ(1-\eta\mu)^{j-1}$ for $j = 1, ..., Q$, where $B \geq \frac{1}{Q(1-\eta\mu)^{Q-1}}$. Then, the bias satisfies*

$$\left\|\mathbb{E}v_Q - [\nabla_y^2 g(x_k, y_k^T)]^{-1} \nabla_y f(x_k, y_k^T)\right\| \leq \mu^{-1}(1-\eta\mu)^{Q+1} M. \tag{8}$$

*Furthermore, the estimation variance is given by*

$$\mathbb{E}\|v_Q - [\nabla_y^2 g(x_k, y_k^T)]^{-1} \nabla_y f(x_k, y_k^T)\|^2 \leq \frac{4\eta^2 L^2 M^2}{\mu^2} \frac{1}{B} + \frac{4(1-\eta\mu)^{2Q+2} M^2}{\mu^2} + \frac{2M^2}{\mu^2 D_f}. \tag{9}$$

Proposition 3 shows that if we choose $Q$ and $B$ at the order level of $\mathcal{O}(\log \frac{1}{\epsilon})$ and $\mathcal{O}(1/\epsilon)$, the bias and variance are smaller than $\mathcal{O}(\epsilon)$, and the required number of samples is $\sum_{j=1}^Q BQ(1-\eta\mu)^{j-1} = \mathcal{O}\left(\epsilon^{-1} \log \frac{1}{\epsilon}\right)$. Note that the chosen batch size $|\mathcal{B}_{Q+1-j}|$ exponentially decays w.r.t. $j$. In comparison, the uniform choice of all $|\mathcal{B}_j|$ would yield a worse complexity of $\mathcal{O}(\epsilon^{-1}(\log \frac{1}{\epsilon})^2)$.

We next analyze stocBiO when the objective function $\Phi(x) := f(x, y^*(x))$ is nonconvex.

**Theorem 3.** *Suppose Assumptions 1, 2, 3 and 4 hold. Define parameter $L_\Phi = L + \frac{2L^2 + \tau M^2}{\mu} + \frac{\rho L M + L^3 + \tau M L}{\mu^2} + \frac{\rho L^2 M}{\mu^3} = \mathcal{O}\left(\kappa^3\right)$, choose stepsize $\beta = \frac{1}{4L_\Phi}$, and set $\eta < \frac{1}{L}$ in Algorithm 3. Set*
$$T \geq \max\left\{ \frac{\log\left(12 + \frac{48\beta^2 L^2}{\mu^2}(L + \frac{L^2}{\mu} + \frac{M\tau}{\mu} + \frac{LM\rho}{\mu^2})^2\right)}{2\log(\frac{L+\mu}{L-\mu})}, \frac{\log\left(\sqrt{\beta}(L + \frac{L^2}{\mu} + \frac{M\tau}{\mu} + \frac{LM\rho}{\mu^2})\right)}{\log(\frac{L+\mu}{L-\mu})} \right\}. \text{ Then, we have}$$

$$\frac{1}{K}\sum_{k=0}^{K-1}\mathbb{E}\|\nabla\Phi(x_k)\|^2 \leq \frac{32L_\Phi(\Phi(x_0) - \inf_x \Phi(x) + \frac{5}{2}\|y_0 - y^*(x_0)\|^2)}{K} + 72\kappa^2 M^2(1 - \eta\mu)^{2Q}$$

$$+ \frac{40\left(L + \frac{L^2}{\mu} + \frac{M\tau}{\mu} + \frac{LM\rho}{\mu^2}\right)^2 \sigma^2}{L\mu}\frac{1}{S} + \frac{16\kappa^2 M^2}{D_g} + \frac{(8 + 32\kappa^2)M^2}{D_f} + \frac{64\kappa^2 M^2}{B}. \tag{10}$$

*In order to achieve an $\epsilon$-accurate stationary point, we have*

- *Gradient complexity: $\text{Gc}(F, \epsilon) = \mathcal{O}(\kappa^5 \epsilon^{-2}), \text{Gc}(G, \epsilon) = \mathcal{O}(\kappa^9 \epsilon^{-2})$.*

- *Jacobian- and Hessian-vector product: $\text{JV}(G, \epsilon) = \mathcal{O}(\kappa^5 \epsilon^{-2}), \text{HV}(G, \epsilon) = \widetilde{\mathcal{O}}(\kappa^6 \epsilon^{-2})$.*

Theorem 3 shows that stocBiO converges sublinearly with the convergence error decaying exponentially w.r.t. $Q$ and sublinearly w.r.t. the batch sizes $S, D_g, D_f$ for gradient estimation and $B$ for Hessian inverse estimation. In addition, it can be seen that the total number $T$ of the inner-loop steps is chosen at nearly a constant level, rather than a typical choice of $\Theta(\log(\frac{1}{\epsilon}))$.

As shown in Table 2, the gradient complexities of our proposed algorithm in terms of $F$ and $G$ improve those of BSA in Ghadimi & Wang (2018) by an order of $\kappa$ and $\epsilon^{-1}$, respectively. In addition, the Jacobian-vector product complexity $\text{JV}(G, \epsilon)$ of our algorithm improves that of BSA by the order of $\kappa$. In terms of the accuracy $\epsilon$, our gradient, Jacobian- and Hessian-vector product complexities improve those of TTSA in Hong et al. (2020) all by an order of $\epsilon^{-0.5}$.

## 5 APPLICATIONS TO META-LEARNING

### 5.1 META-LEARNING WITH COMMON EMBEDDING MODEL

Consider the few-shot meta-learning problem with $m$ tasks $\{\mathcal{T}_i, i = 1, ..., m\}$ sampled from distribution $P_\mathcal{T}$. Each task $\mathcal{T}_i$ has a loss function $\mathcal{L}(\phi, w_i; \xi)$ over each data sample $\xi$, where $\phi$ are the parameters of an embedding model shared by all tasks, and $w_i$ are the task-specific parameters. The goal of this framework is to find good parameters $\phi$ for all tasks, and building on the embedded features, each task then adapts its own parameters $w_i$ by minimizing its loss.

The model training takes a bilevel procedure. In the lower-level stage, building on the embedded features, the base learner of task $\mathcal{T}_i$ searches $w_i^*$ as the minimizer of its loss function over a training set $\mathcal{S}_i$. In the upper-level stage, the meta-learner evaluates the minimizers $w_i^*, i = 1, ..., m$ on held-out test sets, and optimizes $\phi$ of the embedding model over all tasks. Specifically, let $\widetilde{w} = (w_1, ..., w_m)$ denote all task-specific parameters. Then, the objective function is given by

$$\min_\phi \mathcal{L}_\mathcal{D}(\phi, \widetilde{w}^*) := \frac{1}{m}\sum_{i=1}^m \underbrace{\frac{1}{|\mathcal{D}_i|}\sum_{\xi \in \mathcal{D}_i} \mathcal{L}(\phi, w_i^*; \xi)}_{\mathcal{L}_{\mathcal{D}_i}(\phi, w_i^*): \text{ task-specific upper-level loss}}$$

$$\text{s.t. } \widetilde{w}^* = \arg\min_{\widetilde{w}} \mathcal{L}_\mathcal{S}(\phi, \widetilde{w}) = \arg\min_{(w_1, ..., w_m)} \frac{1}{m}\sum_{i=1}^m \Big( \underbrace{\frac{1}{|\mathcal{S}_i|}\sum_{\xi \in \mathcal{S}_i} \mathcal{L}(\phi, w_i; \xi) + \mathcal{R}(w_i)}_{\mathcal{L}_{\mathcal{S}_i}(\phi, w_i): \text{ task-specific lower-level loss}} \Big), \tag{11}$$

where $\mathcal{S}_i$ and $\mathcal{D}_i$ are the training and test datasets of task $\mathcal{T}_i$, and $\mathcal{R}(w_i)$ is a strongly-convex regularizer, e.g., $L^2$. Note that the lower-level problem is equivalent to solving each $w_i^*$ as a minimizer of the task-specific loss $\mathcal{L}_{\mathcal{S}_i}(\phi, w_i)$ for $i = 1, ..., m$. In practice, $w_i$ often corresponds to the parameters of the last *linear* layer of a neural network and $\phi$ are the parameters of the remaining layers (e.g., 4 convolutional layers in Bertinetto et al. (2018); Ji et al. (2020a)), and hence the lower-level function is *strongly-convex* w.r.t. $\widetilde{w}$ and the upper-level function $\mathcal{L}_\mathcal{D}(\phi, \widetilde{w}^*(\phi))$ is generally non-convex w.r.t. $\phi$. In addition, due to the small sizes of datasets $\mathcal{D}_i$ and $\mathcal{S}_i$ in few-shot learning, all

updates for each task $\mathcal{T}_i$ use *full gradient descent* without data resampling. As a result, AID-BiO and ITD-BiO in Algorithm 1 can be applied here. In some applications where the number $m$ of tasks is large, it is more efficient to sample a batch $\mathcal{B}$ of i.i.d. tasks from $\{\mathcal{T}_i, i = 1, ..., m\}$ at each meta (outer) iteration, and optimizes the mini-batch versions $\mathcal{L}_{\mathcal{D}}(\phi, \widetilde{w}; \mathcal{B}) = \frac{1}{|\mathcal{B}|}\sum_{i \in \mathcal{B}} \mathcal{L}_{\mathcal{D}_i}(\phi, w_i)$ and $\mathcal{L}_{\mathcal{S}}(\phi, \widetilde{w}; \mathcal{B}) = \frac{1}{|\mathcal{B}|}\sum_{i \in \mathcal{B}} \mathcal{L}_{\mathcal{S}_i}(\phi, w_i)$ instead. The following theorem provides the convergence analysis of ITD-BiO for this case.

**Theorem 4.** *Suppose Assumptions 1, 2 and 3 hold and suppose each task loss $\mathcal{L}_{\mathcal{S}_i}(\phi, w_i)$ is $\mu$-strongly-convex w.r.t. $w_i$. Choose the same parameters $\beta, T$ as in Theorem 2. Then, we have*

$$\frac{1}{K}\sum_{k=0}^{K-1} \mathbb{E}\|\nabla\Phi(\phi_k)\|^2 \leq \frac{16L_{\Phi}(\Phi(\phi_0) - \inf_{\phi}\Phi(\phi))}{K} + \frac{2\epsilon}{3} + \left(1 + \frac{L}{\mu}\right)^2 \frac{M^2}{8|\mathcal{B}|}.$$

Theorem 4 shows that compared to the full batch (i.e., without task sampling) case in eq. (11), the task sampling introduces a variance term $\mathcal{O}(\frac{1}{|\mathcal{B}|})$ due to the stochastic nature of the algorithm. Using an approach similar to Theorem 4, we can derive a similar result for AID-BiO.

## 5.2 EXPERIMENTS

To validate our theoretical results for deterministic bilevel optimization, we compare the performance among the following four algorithms: ITD-BiO, AID-BiO-constant (AID-BiO with a constant number of inner-loop steps as in our analysis), AID-BiO-increasing (AID-BiO with an increasing number of inner-loop steps under analysis in Ghadimi & Wang (2018)), and two popular meta-learning algorithms MAML[2] (Finn et al., 2017) and ANIL[3] (Raghu et al., 2019). We conduct experiments over a 5-way 5-shot task on two benchmark datasets: FC100 and miniImageNet, and the results are averaged over 10 trials with different random seeds. Due to the space limitations, we provide the model architectures, hyperparameter settings and additional experiments in Appendix B.

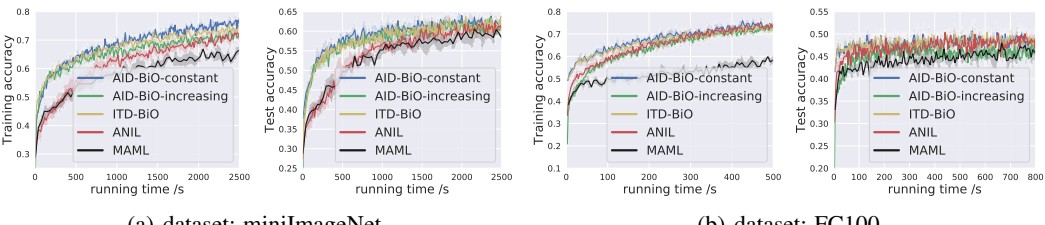

(a) dataset: miniImageNet        (b) dataset: FC100

Figure 1: Convergence of various algorithms on meta-learning. For each dataset, left plot: training accuracy v.s. running time; right plot: test accuracy v.s. running time.

It can be seen from Figure 1 that for both the miniImageNet and FC100 datasets, AID-BiO-constant converges faster than AID-BiO-increasing in terms of both the training accuracy and test accuracy, and achieves a better final test accuracy than ANIL and MAML. This demonstrates the superior improvement of our developed analysis over existing analysis in Ghadimi & Wang (2018) for AID-BiO algorithm. Moreover, it can be observed that AID-BiO is slightly faster than ITD-BiO in terms of the training accuracy and test accuracy. This is also in consistence with our theoretical results.

## 6 CONCLUSION

In this paper, we develop a general and enhanced finite-time analysis for the nonconvex-strongly-convex bilevel deterministic optimization, and propose a novel algorithm for the stochastic setting whose computational complexity outperforms the best known results order-wisely. We also provide the theoretical guarantee of various bilevel optimizers in meta-learning and hyperparameter optimization. The experiments validate our theoretical results and demonstrate the effectiveness of the proposed algorithm. We anticipate that the finite-time analysis that we develop will be useful for analyzing other bilevel optimization problems with different loss geometries, and the proposed algorithms will be useful for other applications such as reinforcement learning and Stackelberg game.

---

[2]MAML consists of an inner loop for task adaptation and an outer loop for meta initialization training.

[3]ANIL refers to almost no inner loop, which is an efficient MAML variant with task-specific adaption on the last-layer of parameters.

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

# Supplementary Materials

## A  APPLICATION TO HYPERPARAMETER OPTIMIZATION

### A.1  HYPERPARAMETER OPTIMIZATION

The goal of hyperparameter optimization (Franceschi et al., 2018; Feurer & Hutter, 2019) is to search for representation or regularization parameters $\lambda$ to minimize the validation error evaluated over the learner's parameters $w^*$, where $w^*$ is the minimizer of the inner-loop regularized training error. Mathematically, the objective function is given by

$$\min_{\lambda} \mathcal{L}_{\mathcal{D}_{\text{val}}}(\lambda) = \frac{1}{|\mathcal{D}_{\text{val}}|} \sum_{\xi \in \mathcal{D}_{\text{val}}} \mathcal{L}(w^*(\lambda); \xi)$$

$$\text{s.t.} \quad w^*(\lambda) = \arg\min_{w} \mathcal{L}_{\mathcal{D}_{\text{tr}}}(w, \lambda) := \frac{1}{|\mathcal{D}_{\text{tr}}|} \sum_{\xi \in \mathcal{D}_{\text{tr}}} \big( \mathcal{L}(w, \lambda; \xi) + \mathcal{R}(w, \lambda) \big), \quad (12)$$

where $\mathcal{D}_{\text{val}}$ and $\mathcal{D}_{\text{tr}}$ are validation and training data, $\mathcal{L}$ is the loss, and $\mathcal{R}(w, \lambda)$ is a regularizer.

In practice, the lower-level function $\mathcal{L}_{\mathcal{D}_{\text{tr}}}(w, \lambda)$ is often strongly-convex w.r.t. $w$. For example, for the data hyper-cleaning application proposed by Franceschi et al. (2018); Shaban et al. (2019), the predictor is modeled by a linear classifier, and the loss function $\mathcal{L}(w; \xi)$ is convex w.r.t. $w$ and $\mathcal{R}(w, \lambda)$ is a strongly-convex regularizer, e.g., $L^2$ regularization. In addition, the sample sizes of $\mathcal{D}_{\text{val}}$ and $\mathcal{D}_{\text{tr}}$ are often large, and stochastic algorithms are preferred for achieving better efficiency. As a result, the above hyperparameter optimization falls into the stochastic bilevel optimization we study in eq. (2), and we can apply the proposed stocBiO algorithm here and Theorem 3 establishes its finite-time performance guarantee.

### A.2  EXPERIMENTS

We compare our proposed stocBiO with the following baseline bilevel optimization algorithms.

- **BSA** (Ghadimi & Wang, 2018): implicit gradient based stochastic bilevel optimizer via single-sample data sampling.
- **TTSA** (Hong et al., 2020): two-time-scale stochastic optimizer via single-sample data sampling.
- **HOAG** (Pedregosa, 2016): a hyperparameter optimization algorithm with approximate gradient. We use the implementation in the repository `https://github.com/fabianp/hoag`.
- **reverse** (Franceschi et al., 2017): an iterative differentiation based method that approximates the hypergradient via backpropagation. We use its implementation in `https://github.com/prolearner/hypertorch`.
- **AID-FP** (Grazzi et al., 2020): AID with the fixed-point method. We use its implementation in `https://github.com/prolearner/hypertorch`
- **AID-CG** (Grazzi et al., 2020): AID with the conjugate gradient method. We use its implementation in `https://github.com/prolearner/hypertorch`.

We demonstrate the effectiveness of the proposed stocBiO algorithm on two experiments: data hyper-cleaning and logistic regression.

**Logistic Regression on 20 Newsgroup:** We compare the performance of our algorithm **stocBiO** with the existing baseline algorithms **reverse, AID-FP, AID-CG and HOAG** over a logistic regression problem on 20 Newsgroup dataset Grazzi et al. (2020). The objective function of such a problem is given by

$$\min_{\lambda} E(\lambda, w^*) = \frac{1}{|\mathcal{D}_{\text{val}}|} \sum_{(x_i, y_i) \in \mathcal{D}_{\text{val}}} L(x_i w^*, y_i)$$

$$\text{s.t.} \quad w^* = \arg\min_{w \in \mathbb{R}^{p \times c}} \Big( \frac{1}{|\mathcal{D}_{\text{tr}}|} \sum_{(x_i, y_i) \in \mathcal{D}_{\text{tr}}} L(x_i w, y_i) + \frac{1}{cp} \sum_{i=1}^{c} \sum_{j=1}^{p} \exp(\lambda_j) w_{ij}^2 \Big),$$

where $L$ is the cross-entropy loss, $c = 20$ is the number of topics, and $p = 101631$ is the feature dimension. Following Grazzi et al. (2020), we use SGD as the optimizer for the outer-loop update for all algorithms. For reverse, AID-FP, AID-CG, we use the suggested and well-tuned hyperparameter setting in their implementations `https://github.com/prolearner/hypertorch` on this application. In specific, they choose the inner- and outer-loop stepsizes as 100, the number of inner loops as 10, the number of CG steps as 10. For HOAG, we use the same parameters as reverse, AID-FP, AID-CG. For stocBiO, we use the same parameters as reverse, AID-FP, AID-CG, and choose $\eta = 0.5, Q = 10$. We use stocBiO-$B$ as a shorthand of stocBiO with a batch size of $B$.

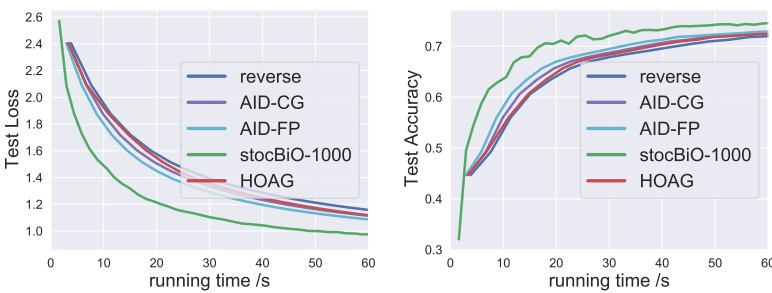

Figure 2: Comparison of various algorithms on logistic regression on 20 Newsgroup dataset. For left plot: test loss v.s. running time; right plot: test accuracy v.s. running time

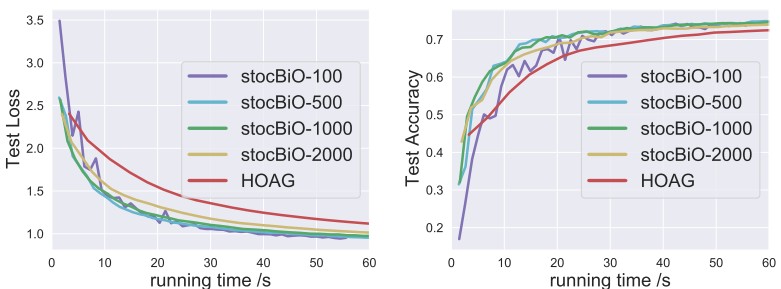

Figure 3: Convergence rate of stocBiO with different batch sizes.

As shown in Figure 2, the proposed stocBiO achieves the fastest convergence rate as well as the best test accuracy among all comparison algorithms. This demonstrates the practical advantage of our proposed algorithm stocBiO. Note that we do not include BSA and TTSA in the comparison, because they converge too slowly with a large variance, and are much worse than the other competing algorithms. In addition, we investigate the impact of the batch size on the performance of our stocBiO in Figure 3. It can be seen that stocBiO outperforms HOAG under the batch sizes of $100, 500, 1000, 2000$. This shows that the performance of stocBiO is not very sensitive to the batch size, and hence the tuning of the batch size is easy to handle in practice.

**Data Hyper-Cleaning on MNIST.** We first compare the performance of our proposed algorithm stocBiO with other baseline algorithms BSA, TTSA, HOAG[4] on a hyperparameter optimization problem: data hyper-cleaning (Shaban et al., 2019) on a dataset derived from MNIST (LeCun et al., 1998), which consists of 20000 images for training, 5000 images for validation, and 10000 images for testing. Data hyper-cleaning is to train a classifier in a corrupted setting where each label of training data is replaced by a random class number with a probability $p$ (i.e., the corruption rate).

---

[4]We do not include reverse, AID-CG and AID-FG because they perform similarly to HOAG.

The objective function is given by

$$\min_\lambda E(\lambda, w^*) = \frac{1}{|\mathcal{D}_{\text{val}}|} \sum_{(x_i, y_i) \in \mathcal{D}_{\text{val}}} L(w^* x_i, y_i)$$

$$\text{s.t.} \quad w^* = \arg\min_w \mathcal{L}(w, \lambda) := \frac{1}{|\mathcal{D}_{\text{tr}}|} \sum_{(x_i, y_i) \in \mathcal{D}_{\text{tr}}} \sigma(\lambda_i) L(w x_i, y_i) + C_r \|w\|^2,$$

where $L$ is the cross-entropy loss, $\sigma(\cdot)$ is the sigmoid function, $C_r$ is a regularization parameter. Following Shaban et al. (2019), we choose $C_r = 0.001$. All results are averaged over 10 trials with different random seeds. We adopt Adam (Kingma & Ba, 2014) as the optimizer for the outer-loop update for all algorithms. For stochastic algorithms, we set the batch size as 50 for stocBiO, and 1 for BSA and TTSA because they use the single-sample data sampling. For all algorithms, we use a grid search to choose the inner-loop stepsize from $\{0.01, 0.1, 1, 10\}$, the outer-loop stepsize from $\{10^i, i = -4, -3, -2, -1, 0, 1, 2, 3, 4\}$, and the number $T$ of inner-loop steps from $\{1, 10, 50, 100, 200, 1000\}$, where values that achieve the lowest loss after a fixed running time are selected. For stocBiO, BSA, and TTSA, we choose $\eta$ from $\{0.5 \times 2^i, i = -3, -2, -1, 0, 1, 2, 3\}$, and $Q$ from $\{3 \times 2^i, i = 0, 1, 2, 3\}$.

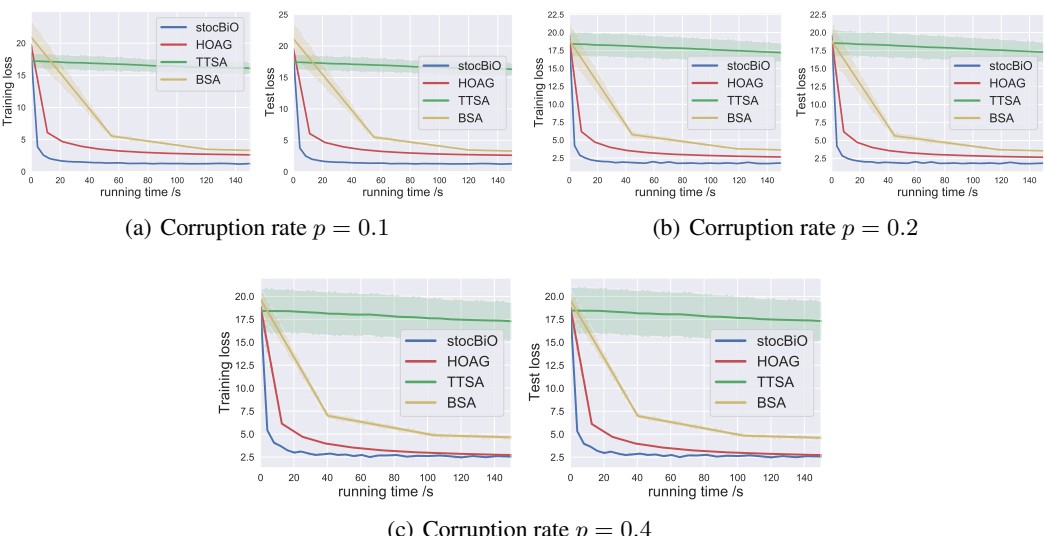

(a) Corruption rate $p = 0.1$    (b) Corruption rate $p = 0.2$

(c) Corruption rate $p = 0.4$

Figure 4: Convergence of various algorithms on hyperparameter optimization at different corruption rates. For each corruption rate $p$, left plot: training loss v.s. running time; right plot: test loss v.s. running time.

It can be seen from Figure 4 that our proposed stocBiO algorithm achieves the fastest convergence rate among all competing algorithms in terms of both the training loss and the test loss. In addition, it is observed that such an improvement is more significant when the corruption rate $p$ is smaller. We note that the stochastic algorithm TTSA converges very slowly with a large variance. This is because TTSA updates the costly outer loop more frequently than other algorithms, and has a larger variance due to the single-sample data sampling. As a comparison, our stocBiO achieves a much lower variance for hypergradient estimation as well as a much faster convergence rate. This verifies our theoretical results in Theorem 3.

# B  FURTHER SPECIFICATIONS ON META-LEARNING EXPERIMENTS

## B.1  DATASETS AND MODEL ARCHITECTURES

FC100 (Oreshkin et al., 2018) is a dataset derived from CIFAR-100 (Krizhevsky & Hinton, 2009), and contains 100 classes with each class consisting of 600 images of size 32. Following Oreshkin et al. (2018), these 100 classes are split into 60 classes for meta-training, 20 classes for meta-validation, and 20 classes for meta-testing. For all comparison algorithms, we use a 4-layer convolutional neural networks (CNN) with four convolutional blocks, in which each convolutional block

contains a $3 \times 3$ convolution (padding $= 1$, stride $= 2$), batch normalization, ReLU activation, and $2 \times 2$ max pooling. Each convolutional layer has $64$ filters.

The miniImageNet dataset (Vinyals et al., 2016) is generated from ImageNet Russakovsky et al. (2015), and consists of 100 classes with each class containing 600 images of size $84 \times 84$. Following the repository Arnold et al. (2019), we partition these classes into 64 classes for meta-training, 16 classes for meta-validation, and 20 classes for meta-testing. Following the repository (Arnold et al., 2019), we use a four-layer CNN with four convolutional blocks, where each block sequentially consists of a $3 \times 3$ convolution, batch normalization, ReLU activation, and $2 \times 2$ max pooling. Each convolutional layer has $32$ filters.

## B.2 IMPLEMENTATIONS AND HYPERPARAMETER SETTINGS

We adopt the existing implementations in the repository (Arnold et al., 2019) for ANIL and MAML. For all algorithms, we adopt Adam (Kingma & Ba, 2014) as the optimizer for the outer-loop update.

**Parameter selection for the experiments in Figure 1(a):** For ANIL and MAML, we adopt the suggested hyperparameter selection in the repository (Arnold et al., 2019). In specific, for ANIL, we choose the inner-loop stepsize as $0.1$, the outer-loop (meta) stepsize as $0.002$, the task sampling size as 32, and the number of inner-loop steps as 5L. For MAML, we choose the inner-loop stepsize as $0.5$, the outer-loop stepsize as $0.003$, the task sampling sizeas 32, and the number of inner-loop steps as 3. For ITD-BiO, AID-BiO-constant and AID-BiO-increasing, we use a grid search to choose the inner-loop stepsize from $\{0.01, 0.1, 1, 10\}$, the task sampling size from $\{32, 128, 256\}$, and the outer-loop stepsize from $\{10^i, i = -3, -2, -1, 0, 1, 2, 3\}$, where values that achieve the lowest loss after a fixed running time are selected. For ITD-BiO and AID-BiO-constant, we choose the number of inner-loop steps from $\{5, 10, 15, 20, 50\}$, and for AID-BiO-increasing, we choose the number of inner-loop steps as $\lceil c(k + 1)^{1/4} \rceil$ as adopted by the analysis in Ghadimi & Wang (2018), where we choose $c$ from $\{0.5, 2, 5, 10, 50\}$. For both AID-BiO-constant and AID-BiO-increasing, we choose the number $N$ of CG steps for solving the linear system from $\{5, 10, 15\}$.

**Parameter selection for the experiments in Figure 1(b):** For ANIL and MAML, we adopt the suggested hyperparameter selection in the repository (Arnold et al., 2019). Specifically, for ANIL, we choose the inner-loop stepsize as $0.1$, the outer-loop (meta) stepsize as $0.001$, the task sampling size as 32 and the number of inner-loop steps as 10. For MAML, we choose the inner-loop stepsize as $0.5$, the outer-loop stepsize as $0.001$, the task samling size as 32, and the number of inner-loop steps as 3. For ITD-BiO, AID-BiO-constant and AID-BiO-increasing, we adopt the same procedure as in the experiments in Figure 1(a).

## B.3 ADDITIONAL RESULTS FOR META LEARNING

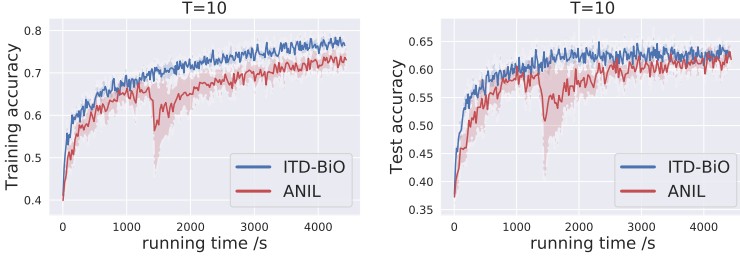

Figure 5: Comparison of ITD-BiO and ANIL on miniImageNet dataset with $T = 10$.

In this subsection, we compare the robustness between bilevel optimizer ITD-BiO (AID-BiO performs similarly to ITD-BiO in terms of the convergence rate) and ANIL (ANIL outperforms MAML in general) to the number of inner-loop steps. For the experiments in Figure 5, we choose the inner-loop stepsize as $0.05$, the outer-loop (meta) stepsize as $0.002$, the mini-batch size as 32, and the number $T$ of inner-loop steps as 10 for both ANIL and ITD-BiO. For the experiments in Figure 6, we choose the inner-loop stepsize as $0.1$, the outer-loop (meta) stepsize as $0.001$, the mini-batch size as 32, and the number $T$ of inner-loop steps as 20 for both ANIL and ITD-BiO.

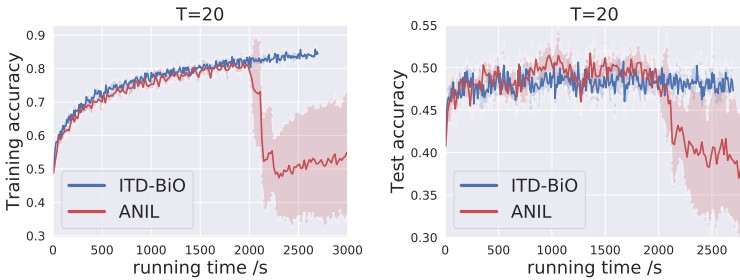

Figure 6: Comparison of ITD-BiO and ANIL on FC100 dataset with $T = 20$.

It can be seen from Figure 5 and Figure 6 that when the number of inner-loop steps become larger, i.e., $T = 10$ for miniImageNet and $T = 20$ for FC100, the bilevel optimizer ITD-BiO converges stably with a small variance, whereas ANIL suffers from a sudden descent at 1500s on miniImageNet and even diverges after 2000s on FC100.

## C   SUPPORTING LEMMAS

In this section, we provide some auxiliary lemmas used for proving the main convergence results.

First note that the Lipschitz properties in Assumption 2 imply the following lemma.

**Lemma 1.** *Suppose Assumption 2 holds. Then, the stochastic derivatives $\nabla F(z; \xi)$, $\nabla G(z; \xi)$, $\nabla_x \nabla_y G(z; \xi)$ and $\nabla_y^2 G(z; \xi)$ have bounded variances, i.e., for any $z$ and $\xi$,*

- $\mathbb{E}_\xi \|\nabla F(z; \xi) - \nabla f(z)\|^2 \leq M^2$.

- $\mathbb{E}_\xi \|\nabla_x \nabla_y G(z; \xi) - \nabla_x \nabla_y g(z)\|^2 \leq L^2$.

- $\mathbb{E}_\xi \|\nabla_y^2 G(z; \xi) - \nabla_y^2 g(z)\|^2 \leq L^2$.

Recall that $\Phi(x) = f(x, y^*(x))$ in eq. (2). Then, we use the following lemma to characterize the Lipschitz properties of $\nabla \Phi(x)$, which is adapted from Lemma 2.2 in Ghadimi & Wang (2018).

**Lemma 2.** *Suppose Assumptions 1, 2 and 3 hold. Then, we have, for any $x, x' \in \mathbb{R}^p$,*

$$\|\nabla \Phi(x) - \nabla \Phi(x')\| \leq L_\Phi \|x - x'\|,$$

*where the constant $L_\Phi$ is given by*

$$L_\Phi = L + \frac{2L^2 + \tau M^2}{\mu} + \frac{\rho LM + L^3 + \tau ML}{\mu^2} + \frac{\rho L^2 M}{\mu^3}. \tag{13}$$

## D   PROOF OF PROPOSITIONS IN SECTION 2

In this section, we provide the proofs for Proposition 1 and Proposition 2 in Section 2.

### D.1   PROOF OF PROPOSITION 1

Using the chain rule over the gradient $\nabla \Phi(x_k) = \frac{\partial f(x_k, y^*(x_k))}{\partial x_k}$, we have

$$\nabla \Phi(x_k) = \nabla_x f(x_k, y^*(x_k)) + \frac{\partial y^*(x_k)}{\partial x_k} \nabla_y f(x_k, y^*(x_k)). \tag{14}$$

Based on the optimality of $y^*(x_k)$, we have $\nabla_y g(x_k, y^*(x_k)) = 0$, which, using the implicit differentiation w.r.t. $x_k$, yields

$$\nabla_x \nabla_y g(x_k, y^*(x_k)) + \frac{\partial y^*(x_k)}{\partial x_k} \nabla_y^2 g(x_k, y^*(x_k)) = 0. \tag{15}$$

Let $v_k^*$ be the solution of the linear system $\nabla_y^2 g(x_k, y^*(x_k))v = \nabla_y f(x_k, y^*(x_k))$. Then, multiplying $v_k^*$ at the both sides of eq. (15), yields

$$-\nabla_x \nabla_y g(x_k, y^*(x_k))v_k^* = \frac{\partial y^*(x_k)}{\partial x_k}\nabla_y^2 g(x_k, y^*(x_k))v_k^* = \frac{\partial y^*(x_k)}{\partial x_k}\nabla_y f(x_k, y^*(x_k)),$$

which, in conjunction with eq. (14), yields the proof.

### D.2   PROOF OF PROPOSITION 2

Based on the iterative update of line 5 in Algorithm 1, we have $y_k^T = y_k^0 - \alpha \sum_{t=0}^{T-1} \nabla_y g(x_k, y_k^t)$, which, combined with the fact that $\nabla_y g(x_k, y_k^t)$ is differentiable w.r.t. $x_k$, indicates that the inner output $y_k^T$ is differentiable w.r.t. $x_k$. Then, based on the chain rule, we have

$$\frac{\partial f(x_k, y_k^T)}{\partial x_k} = \nabla_x f(x_k, y_k^T) + \frac{\partial y_k^T}{\partial x_k}\nabla_y f(x_k, y_k^T). \tag{16}$$

Based on the iterative updates that $y_k^t = y_k^{t-1} - \alpha \nabla_y g(x_k, y_k^{t-1})$ for $t = 1, ..., T$, we have

$$\begin{aligned}
\frac{\partial y_k^t}{\partial x_k} &= \frac{\partial y_k^{t-1}}{\partial x_k} - \alpha \nabla_x \nabla_y g(x_k, y_k^{t-1}) - \alpha \frac{\partial y_k^{t-1}}{\partial x_k}\nabla_y^2 g(x_k, y_k^{t-1}) \\
&= \frac{\partial y_k^{t-1}}{\partial x_k}(I - \alpha \nabla_y^2 g(x_k, y_k^{t-1})) - \alpha \nabla_x \nabla_y g(x_k, y_k^{t-1}).
\end{aligned}$$

Telescoping the above equality over $t$ from 1 to $T$ yields

$$\begin{aligned}
\frac{\partial y_k^T}{\partial x_k} &= \frac{\partial y_k^0}{\partial x_k}\prod_{t=0}^{T-1}(I - \alpha \nabla_y^2 g(x_k, y_k^t)) - \alpha \sum_{t=0}^{T-1}\nabla_x \nabla_y g(x_k, y_k^t)\prod_{j=t+1}^{T-1}(I - \alpha \nabla_y^2 g(x_k, y_k^j)) \\
&\overset{(i)}{=} -\alpha \sum_{t=0}^{T-1}\nabla_x \nabla_y g(x_k, y_k^t)\prod_{j=t+1}^{T-1}(I - \alpha \nabla_y^2 g(x_k, y_k^j)).
\end{aligned} \tag{17}$$

where $(i)$ follows from the fact that $\frac{\partial y_k^0}{\partial x_k} = 0$. Combining eq. (16) and eq. (17) finishes the proof.

## E   CONVERGENCE PROOFS FOR AID-BIO IN SECTION 4.1

For notation simplification, we define the following quantities.

$$\begin{aligned}
\Gamma &= 3L^2 + \frac{3\tau^2 M^2}{\mu^2} + 6L^2\big(1 + \sqrt{\kappa}\big)^2\big(\kappa + \frac{\rho M}{\mu^2}\big)^2, \quad \delta_{T,N} = \Gamma(1 - \alpha\mu)^T + 6L^2\kappa\big(\frac{\sqrt{\kappa} - 1}{\sqrt{\kappa} + 1}\big)^{2N} \\
\Omega &= 8\Big(\beta\kappa^2 + \frac{2\beta ML}{\mu^2} + \frac{2\beta LM\kappa}{\mu^2}\Big)^2, \quad \Delta_0 = \|y_0 - y^*(x_0)\|^2 + \|v_0^* - v_0\|^2.
\end{aligned} \tag{18}$$

We first provide some supporting lemmas. The following lemma characterizes the Hypergradient estimation error $\|\widehat{\nabla}\Phi(x_k) - \nabla\Phi(x_k)\|$, where $\widehat{\nabla}\Phi(x_k)$ is given by eq. (3) via implicit differentiation.

**Lemma 3.** *Suppose Assumptions 1, 2 and 3 hold. Then, we have*

$$\|\widehat{\nabla}\Phi(x_k) - \nabla\Phi(x_k)\|^2 \leq \Gamma(1 - \alpha\mu)^T\|y^*(x_k) - y_k^0\|^2 + 6L^2\kappa\big(\frac{\sqrt{\kappa} - 1}{\sqrt{\kappa} + 1}\big)^{2N}\|v_k^* - v_k^0\|^2.$$

*where $\Gamma$ is given by eq. (18).*

**Proof of Lemma 3.** Based on the form of $\nabla\Phi(x_k)$ given by Proposition 1, we have

$$\begin{aligned}
\|\widehat{\nabla}\Phi(x_k) - \nabla\Phi(x_k)\|^2 \leq &3\|\nabla_x f(x_k, y^*(x_k)) - \nabla_x f(x_k, y_k^T)\|^2 + 3\|\nabla_x \nabla_y g(x_k, y_k^T)\|^2\|v_k^* - v_k^N\|^2 \\
&+ 3\|\nabla_x \nabla_y g(x_k, y^*(x_k)) - \nabla_x \nabla_y g(x_k, y_k^T)\|^2\|v_k^*\|^2,
\end{aligned}$$

which, in conjunction with Assumptions 1, 2 and 3, yields

$$\|\widehat{\nabla}\Phi(x_k) - \nabla\Phi(x_k)\|^2 \le 3L^2\|y^*(x_k) - y_k^T\|^2 + 3L^2\|v_k^* - v_k^N\|^2 + 3\tau^2\|v_k^*\|^2\|y_k^T - y^*(x_k)\|^2$$

$$\overset{(i)}{\le} 3L^2\|y^*(x_k) - y_k^T\|^2 + 3L^2\|v_k^* - v_k^N\|^2 + \frac{3\tau^2 M^2}{\mu^2}\|y_k^T - y^*(x_k)\|^2. \quad (19)$$

where $(i)$ follows from the fact that $\|v_k^*\| \le \|(\nabla_y^2 g(x_k, y^*(x_k)))^{-1}\|\|\nabla_y f(x_k, y^*(x_k))\| \le \frac{M}{\mu}$.

For notation simplification, let $\widehat{v}_k = (\nabla_y^2 g(x_k, y_k^T))^{-1}\nabla_y f(x_k, y_k^T)$. We next upper-bound $\|v_k^* - v_k^N\|$ in eq. (19). Based on the convergence result of CG for the quadratic programing, e.g., eq. (17) in Grazzi et al. (2020), we have $\|v_k^N - \widehat{v}_k\| \le \sqrt{\kappa}\left(\frac{\sqrt{\kappa}-1}{\sqrt{\kappa}+1}\right)^N\|v_k^0 - \widehat{v}_k\|$. Based on this inequality, we further have

$$\|v_k^* - v_k^N\| \le \|v_k^* - \widehat{v}_k\| + \|v_k^N - \widehat{v}_k\| \le \|v_k^* - \widehat{v}_k\| + \sqrt{\kappa}\left(\frac{\sqrt{\kappa}-1}{\sqrt{\kappa}+1}\right)^N\|v_k^0 - \widehat{v}_k\|$$

$$\le \left(1 + \sqrt{\kappa}\left(\frac{\sqrt{\kappa}-1}{\sqrt{\kappa}+1}\right)^N\right)\|v_k^* - \widehat{v}_k\| + \sqrt{\kappa}\left(\frac{\sqrt{\kappa}-1}{\sqrt{\kappa}+1}\right)^N\|v_k^* - v_k^0\|. \quad (20)$$

Next, based on the definitions of $v_k^*$ and $\widehat{v}_k$, we have

$$\|v_k^* - \widehat{v}_k\| = \|(\nabla_y^2 g(x_k, y_k^T))^{-1}\nabla_y f(x_k, y_k^T) - (\nabla_y^2 g(x_k, y^*(x_k))^{-1}\nabla_y f(x_k, y^*(x_k))\|$$

$$\le \left(\kappa + \frac{\rho M}{\mu^2}\right)\|y_k^T - y^*(x_k)\|. \quad (21)$$

Combining eq. (19), eq. (20), eq. (21) yields

$$\|\widehat{\nabla}\Phi(x_k) - \nabla\Phi(x_k)\|^2 \le \left(3L^2 + \frac{3\tau^2 M^2}{\mu^2}\right)\|y^*(x_k) - y_k^T\|^2 + 6L^2\kappa\left(\frac{\sqrt{\kappa}-1}{\sqrt{\kappa}+1}\right)^{2N}\|v_k^* - v_k^0\|^2$$

$$+ 6L^2\left(1 + \sqrt{\kappa}\left(\frac{\sqrt{\kappa}-1}{\sqrt{\kappa}+1}\right)^N\right)^2\left(\kappa + \frac{\rho M}{\mu^2}\right)^2\|y_k^T - y^*(x_k)\|^2,$$

which, in conjunction with $\|y_k^T - y^*(x_k)\| \le (1-\alpha\mu)^{\frac{T}{2}}\|y_k^0 - y^*(x_k)\|$ and the notations in eq. (18), finishes the proof. $\square$

**Lemma 4.** *Suppose Assumptions 1, 2 and 3 hold. Choose*

$$T \ge \log\left(36\kappa(\kappa + \frac{\rho M}{\mu^2})^2 + 16(\kappa^2 + \frac{4LM\kappa}{\mu^2})^2\beta^2\Gamma\right)/\log\frac{1}{1-\alpha} = \Theta(\kappa)$$

$$N \ge \frac{1}{2}\log(8\kappa + 48(\kappa^2 + \frac{2ML}{\mu^2} + \frac{2LM\kappa}{\mu^2})^2\beta^2 L^2\kappa)/\log\frac{\sqrt{\kappa}+1}{\sqrt{\kappa}-1} = \Theta(\sqrt{\kappa}), \quad (22)$$

*where $\Gamma$ is given by eq. (18). Then, we have*

$$\|y_k^0 - y^*(x_k)\|^2 + \|v_k^* - v_k^0\|^2 \le \left(\frac{1}{2}\right)^k\Delta_0 + \Omega\sum_{j=0}^{k-1}\left(\frac{1}{2}\right)^{k-1-j}\|\nabla\Phi(x_j)\|^2, \quad (23)$$

*where $\Omega$ and $\Delta_0$ are given by eq. (18).*

**Proof of Lemma 4.** Recall that $y_k^0 = y_{k-1}^T$. Then, we have

$$\|y_k^0 - y^*(x_k)\|^2 \le 2\|y_{k-1}^T - y^*(x_{k-1})\|^2 + 2\|y^*(x_k) - y^*(x_{k-1})\|^2$$

$$\overset{(i)}{\le} 2(1-\alpha\mu)^T\|y_{k-1}^0 - y^*(x_{k-1})\|^2 + 2\kappa^2\beta^2\|\widehat{\nabla}\Phi(x_{k-1})\|^2$$

$$\le 2(1-\alpha\mu)^T\|y_{k-1}^0 - y^*(x_{k-1})\|^2 + 4\kappa^2\beta^2\|\nabla\Phi(x_{k-1}) - \widehat{\nabla}\Phi(x_{k-1})\|^2$$

$$+ 4\kappa^2\beta^2\|\nabla\Phi(x_{k-1})\|^2$$

$$\overset{(ii)}{\le} \left(2(1-\alpha\mu)^T + 4\kappa^2\beta^2\Gamma(1-\alpha\mu)^T\right)\|y^*(x_{k-1}) - y_{k-1}^0\|^2$$

$$+ 24\kappa^4 L^2\beta^2\left(\frac{\sqrt{\kappa}-1}{\sqrt{\kappa}+1}\right)^{2N}\|v_{k-1}^* - v_{k-1}^0\|^2 + 4\kappa^2\beta^2\|\nabla\Phi(x_{k-1})\|^2. \quad (24)$$

where $(i)$ follows from Lemma 2.2 in Ghadimi & Wang (2018) and $(ii)$ follows from Lemma 3. In addition, note that

$$
\begin{aligned}
\|v_k^* - v_k^0\|^2 = \|v_k^* - v_{k-1}^N\|^2 &\leq 2\|v_{k-1}^* - v_{k-1}^N\|^2 + 2\|v_k^* - v_{k-1}^*\|^2 \\
&\overset{(i)}{\leq} 4\left(1 + \sqrt{\kappa}\right)^2\left(\kappa + \frac{\rho M}{\mu^2}\right)^2(1 - \alpha\mu)^T\|y_{k-1}^0 - y^*(x_{k-1})\|^2 \\
&\quad + 4\kappa\left(\frac{\sqrt{\kappa} - 1}{\sqrt{\kappa} + 1}\right)^{2N}\|v_{k-1}^* - v_{k-1}^0\|^2 + 2\|v_k^* - v_{k-1}^*\|^2,
\end{aligned}
\tag{25}
$$

where $(i)$ follows from eq. (20). Combining eq. (25) with $\|v_k^* - v_{k-1}^*\| \leq (\kappa^2 + \frac{2ML}{\mu^2} + \frac{2LM\kappa}{\mu^2})\|x_k - x_{k-1}\|$, we have

$$
\begin{aligned}
\|v_k^* - v_k^0\|^2 \overset{(i)}{\leq} &\left(16\kappa\left(\kappa + \frac{\rho M}{\mu^2}\right)^2 + 4\left(\kappa^2 + \frac{4LM\kappa}{\mu^2}\right)^2\beta^2\Gamma\right)(1 - \alpha\mu)^T\|y_{k-1}^0 - y^*(x_{k-1})\|^2 \\
&+ \left(4\kappa + 48\left(\kappa^2 + \frac{2ML}{\mu^2} + \frac{2LM\kappa}{\mu^2}\right)^2\beta^2 L^2\kappa\right)\left(\frac{\sqrt{\kappa} - 1}{\sqrt{\kappa} + 1}\right)^{2N}\|v_{k-1}^* - v_{k-1}^0\|^2 \\
&+ 4\left(\kappa^2 + \frac{2ML}{\mu^2} + \frac{2LM\kappa}{\mu^2}\right)^2\beta^2\|\nabla\Phi(x_{k-1})\|^2,
\end{aligned}
\tag{26}
$$

where $(i)$ follows from Lemma 3. Combining eq. (24) and eq. (26) yields

$$
\begin{aligned}
\|y_k^0 - &y^*(x_k)\|^2 + \|v_k^* - v_k^0\|^2 \\
\leq &\left(18\kappa\left(\kappa + \frac{\rho M}{\mu^2}\right)^2 + 8\left(\kappa^2 + \frac{4LM\kappa}{\mu^2}\right)^2\beta^2\Gamma\right)(1 - \alpha\mu)^T\|y_{k-1}^0 - y^*(x_{k-1})\|^2 \\
&+ \left(4\kappa + 24\left(\kappa^2 + \frac{2ML}{\mu^2} + \frac{2LM\kappa}{\mu^2}\right)^2\beta^2 L^2\kappa\right)\left(\frac{\sqrt{\kappa} - 1}{\sqrt{\kappa} + 1}\right)^{2N}\|v_{k-1}^* - v_{k-1}^0\|^2 \\
&+ 8\left(\kappa^2 + \frac{2ML}{\mu^2} + \frac{2LM\kappa}{\mu^2}\right)^2\beta^2\|\nabla\Phi(x_{k-1})\|^2,
\end{aligned}
$$

which, in conjunction with eq. (22),

$$
\begin{aligned}
\|y_k^0 - y^*(x_k)\|^2 + \|v_k^* - v_k^0\|^2 \leq &\frac{1}{2}(\|y_{k-1}^0 - y^*(x_{k-1})\|^2 + \|v_{k-1}^* - v_{k-1}^0\|^2) \\
&+ 8\left(\beta\kappa^2 + \frac{2\beta ML}{\mu^2} + \frac{2\beta LM\kappa}{\mu^2}\right)^2\|\nabla\Phi(x_{k-1})\|^2.
\end{aligned}
\tag{27}
$$

Telescoping eq. (27) over $k$ and using the notations in eq. (18), we finish the proof. $\qquad\square$

**Lemma 5.** *Under the same setting as in Lemma 4, we have*

$$
\|\widehat{\nabla}\Phi(x_k) - \nabla\Phi(x_k)\|^2 \leq \delta_{T,N}\left(\frac{1}{2}\right)^k\Delta_0 + \delta_{T,N}\Omega\sum_{j=0}^{k-1}\left(\frac{1}{2}\right)^{k-1-j}\|\nabla\Phi(x_j)\|^2.
$$

*where $\delta_{T,N}$, $\Omega$ and $\Delta_0$ are given by eq. (18).*

**Proof of Lemma 5.** Based on Lemma 3, eq. (18) and using $ab + cd \leq (a+c)(b+d)$ for any positive $a, b, c, d$, we have

$$
\|\widehat{\nabla}\Phi(x_k) - \nabla\Phi(x_k)\|^2 \leq \delta_{T,N}(\|y^*(x_k) - y_k^0\|^2 + \|v_k^* - v_k^0\|^2),
$$

which, in conjunction with Lemma 4, finishes the proof. $\qquad\square$

### E.1 Proof of Theorem 1

In this subsection, provide the proof for Theorem 1 based on the supporting Lemma 5. Based on the smoothness of the function $\Phi(x)$ established in Lemma 2, we have

$$
\begin{aligned}
\Phi(x_{k+1}) \leq & \Phi(x_k) + \langle \nabla \Phi(x_k), x_{k+1} - x_k \rangle + \frac{L_\Phi}{2} \| x_{k+1} - x_k \|^2 \\
\leq & \Phi(x_k) - \beta \langle \nabla \Phi(x_k), \widehat{\nabla} \Phi(x_k) - \nabla \Phi(x_k) \rangle - \beta \| \nabla \Phi(x_k) \|^2 + \beta^2 L_\Phi \| \nabla \Phi(x_k) \|^2 \\
& + \beta^2 L_\Phi \| \nabla \Phi(x_k) - \widehat{\nabla} \Phi(x_k) \|^2 \\
\leq & \Phi(x_k) - \left( \frac{\beta}{2} - \beta^2 L_\Phi \right) \| \nabla \Phi(x_k) \|^2 + \left( \frac{\beta}{2} + \beta^2 L_\Phi \right) \| \nabla \Phi(x_k) - \widehat{\nabla} \Phi(x_k) \|^2, \quad (28)
\end{aligned}
$$

which, combined with Lemma 5, yields

$$
\begin{aligned}
\Phi(x_{k+1}) \leq & \Phi(x_k) - \left( \frac{\beta}{2} - \beta^2 L_\Phi \right) \| \nabla \Phi(x_k) \|^2 + \left( \frac{\beta}{2} + \beta^2 L_\Phi \right) \delta_{T,N} \left( \frac{1}{2} \right)^k \Delta_0 \\
& + \left( \frac{\beta}{2} + \beta^2 L_\Phi \right) \delta_{T,N} \Omega \sum_{j=0}^{k-1} \left( \frac{1}{2} \right)^{k-1-j} \| \nabla \Phi(x_j) \|^2. \quad (29)
\end{aligned}
$$

Telescoping eq. (29) over k from 0 to $K - 1$ yields

$$
\begin{aligned}
\left( \frac{\beta}{2} - \beta^2 L_\Phi \right) \sum_{k=0}^{K-1} \| \nabla \Phi(x_k) \|^2 \leq & \Phi(x_0) - \inf_x \Phi(x) + \left( \frac{\beta}{2} + \beta^2 L_\Phi \right) \delta_{T,N} \Delta_0 \\
& + \left( \frac{\beta}{2} + \beta^2 L_\Phi \right) \delta_{T,N} \Omega \sum_{k=1}^{K-1} \sum_{j=0}^{k-1} \left( \frac{1}{2} \right)^{k-1-j} \| \nabla \Phi(x_j) \|^2,
\end{aligned}
$$

which, using the fact that $\sum_{k=1}^{K-1} \sum_{j=0}^{k-1} \left( \frac{1}{2} \right)^{k-1-j} \| \nabla \Phi(x_j) \|^2 \leq \sum_{k=0}^{K-1} \frac{1}{2^k} \sum_{k=0}^{K-1} \| \nabla \Phi(x_k) \|^2 \leq 2 \sum_{k=0}^{K-1} \| \nabla \Phi(x_k) \|^2$, yields

$$
\begin{aligned}
\left( \frac{\beta}{2} - \beta^2 L_\Phi - \left( \beta \Omega + 2\Omega \beta^2 L_\Phi \right) \delta_{T,N} \right) \sum_{k=0}^{K-1} \| \nabla \Phi(x_k) \|^2 & \\
\leq \Phi(x_0) - \inf_x \Phi(x) + \left( \frac{\beta}{2} + \beta^2 L_\Phi \right) \delta_{T,N} \Delta_0. & \quad (30)
\end{aligned}
$$

Choose $N$ and $T$ such that

$$
\left( \Omega + 2\Omega \beta L_\Phi \right) \delta_{T,N} \leq \frac{1}{4}, \quad \delta_{T,N} \leq 1. \quad (31)
$$

Note that based on the definition of $\delta_{T,N}$ in eq. (18), it suffices to choose $T \geq \Theta(\kappa)$ and $N \geq \Theta(\sqrt{\kappa})$ to satisfy eq. (31). Then, substituting eq. (31) into eq. (30) yields

$$
\left( \frac{\beta}{4} - \beta^2 L_\Phi \right) \sum_{k=0}^{K-1} \| \nabla \Phi(x_k) \|^2 \leq \Phi(x_0) - \inf_x \Phi(x) + \left( \frac{\beta}{2} + \beta^2 L_\Phi \right) \Delta_0, \quad (32)
$$

which, in conjunction with $\beta \leq \frac{1}{8 L_\Phi}$, yields

$$
\frac{1}{K} \sum_{k=0}^{K-1} \| \nabla \Phi(x_k) \|^2 \leq \frac{64 L_\Phi (\Phi(x_0) - \inf_x \Phi(x)) + 5\Delta_0}{K}. \quad (33)
$$

In order to achieve an $\epsilon$-accurate stationary point, we obtain from eq. (33) that AID-BiO requires at most the total number $K = \mathcal{O}(\kappa^3 \epsilon^{-1})$ of outer iterations. Then, based on eq. (3), we have the following complexity results.

- Gradient complexity:
$$
\text{Gc}(f, \epsilon) = 2K = \mathcal{O}(\kappa^3 \epsilon^{-1}), \text{Gc}(g, \epsilon) = KT = \mathcal{O}(\kappa^4 \epsilon^{-1}).
$$

- Jacobian- and Hessian-vector product complexities:
$$
\text{JV}(g, \epsilon) = K = \mathcal{O}\left( \kappa^3 \epsilon^{-1} \right), \text{HV}(g, \epsilon) = KN = \mathcal{O}\left( \kappa^{3.5} \epsilon^{-1} \right).
$$

Then, the proof is complete.

## F    CONVERGENCE PROOFS FOR ITD-BIO IN SECTION 4.1

We first characterize an important estimation property of the outer-loop gradient estimator $\frac{\partial f(x_k, y_k^T)}{\partial x_k}$ in ITD-BiO for approximating the true gradient $\nabla\Phi(x_k)$ based on Proposition 2.

**Lemma 6.** *Suppose Assumptions 1, 2 and 3 hold. Choose $\alpha \leq \frac{1}{L}$. Then, we have*

$$\left\|\frac{\partial f(x_k, y_k^T)}{\partial x_k} - \nabla\Phi(x_k)\right\| \leq \left(\frac{L(L+\mu)(1-\alpha\mu)^{\frac{T}{2}}}{\mu} + \frac{2M(\tau\mu + L\rho)}{\mu^2}(1-\alpha\mu)^{\frac{T-1}{2}}\right)\|y_k^0 - y^*(x_k)\|$$
$$+ \frac{LM(1-\alpha\mu)^T}{\mu}.$$

Lemma 6 shows that the gradient estimation error $\left\|\frac{\partial f(x_k, y_k^T)}{\partial x_k} - \nabla\Phi(x_k)\right\|$ decays exponentially w.r.t. the number $T$ of the inner-loop steps. We note that Grazzi et al. (2020) proved a similar result via a fixed point based approach. As a comparison, our proof of Lemma 6 directly characterizes the rate of the sequence $\left(\frac{\partial y_k^t}{\partial x_k}, t = 0, ..., T\right)$ converging to $\frac{\partial y^*(x_k)}{\partial x_k}$ via the differentiation over all corresponding points along the inner-loop GD path as well as the optimality of the point $y^*(x_k)$.

**Proof of Lemma 6.** Using $\nabla\Phi(x_k) = \nabla_x f(x_k, y^*(x_k)) + \frac{\partial y^*(x_k)}{\partial x_k}\nabla_y f(x_k, y^*(x_k))$ and eq. (16), and using the triangle inequality, we have

$$\left\|\frac{\partial f(x_k, y_k^T)}{\partial x_k} - \nabla\Phi(x_k)\right\|$$

$$=\|\nabla_x f(x_k, y_k^T) - \nabla_x f(x_k, y^*(x_k))\| + \left\|\frac{\partial y_k^T}{\partial x_k} - \frac{\partial y^*(x_k)}{\partial x_k}\right\|\|\nabla_y f(x_k, y_k^T)\|$$

$$+ \left\|\frac{\partial y^*(x_k)}{\partial x_k}\right\|\|\nabla_y f(x_k, y_k^T) - \nabla_y f(x_k, y^*(x_k))\|$$

$$\overset{(i)}{\leq} L\|y_k^T - y^*(x_k)\| + M\left\|\frac{\partial y_k^T}{\partial x_k} - \frac{\partial y^*(x_k)}{\partial x_k}\right\| + L\left\|\frac{\partial y^*(x_k)}{\partial x_k}\right\|\|y_k^T - y^*(x_k)\|, \quad (34)$$

where $(i)$ follows from Assumption 2. Our next step is to upper-bound $\left\|\frac{\partial y_k^T}{\partial x_k} - \frac{\partial y^*(x_k)}{\partial x_k}\right\|$ in eq. (34).

Based on the updates $y_k^t = y_k^{t-1} - \alpha\nabla_y g(x_k, y_k^{t-1})$ for $t = 1, ..., T$ in ITD-BiO and using the chain rule, we have

$$\frac{\partial y_k^t}{\partial x_k} = \frac{\partial y_k^{t-1}}{\partial x_k} - \alpha\left(\nabla_x\nabla_y g(x_k, y_k^{t-1}) + \frac{\partial y_k^{t-1}}{\partial x_k}\nabla_y^2 g(x_k, y_k^{t-1})\right). \quad (35)$$

Based on the optimality of $y^*(x_k)$, we have $\nabla_y g(x_k, y^*(x_k)) = 0$, which, in conjunction with the implicit differentiation theorem, yields

$$\nabla_x\nabla_y g(x_k, y^*(x_k)) + \frac{\partial y^*(x_k)}{\partial x_k}\nabla_y^2 g(x_k, y^*(x_k)) = 0. \quad (36)$$

Substituting eq. (36) into eq. (35) yields

$$\frac{\partial y_k^t}{\partial x_k} - \frac{\partial y^*(x_k)}{\partial x_k}$$

$$=\frac{\partial y_k^{t-1}}{\partial x_k} - \frac{\partial y^*(x_k)}{\partial x_k} - \alpha\left(\nabla_x\nabla_y g(x_k, y_k^{t-1}) + \frac{\partial y_k^{t-1}}{\partial x_k}\nabla_y^2 g(x_k, y_k^{t-1})\right)$$

$$+ \alpha\left(\nabla_x\nabla_y g(x_k, y^*(x_k)) + \frac{\partial y^*(x_k)}{\partial x_k}\nabla_y^2 g(x_k, y^*(x_k))\right)$$

$$=\frac{\partial y_k^{t-1}}{\partial x_k} - \frac{\partial y^*(x_k)}{\partial x_k} - \alpha\left(\nabla_x\nabla_y g(x_k, y_k^{t-1}) - \nabla_x\nabla_y g(x_k, y^*(x_k))\right)$$

$$- \alpha\left(\frac{\partial y_k^{t-1}}{\partial x_k} - \frac{\partial y^*(x_k)}{\partial x_k}\right)\nabla_y^2 g(x_k, y_k^{t-1})$$

$$+ \alpha\frac{\partial y^*(x_k)}{\partial x_k}\left(\nabla_y^2 g(x_k, y^*(x_k)) - \nabla_y^2 g(x_k, y_k^{t-1})\right). \quad (37)$$

Combining eq. (36) and Assumption 2 yields

$$\left\|\frac{\partial y^*(x_k)}{\partial x_k}\right\| = \left\|\nabla_x \nabla_y g(x_k, y^*(x_k)) \left[\nabla_y^2 g(x_k, y^*(x_k))\right]^{-1}\right\| \leq \frac{L}{\mu}. \tag{38}$$

Then, combining eq. (37) and eq. (38) yields

$$\left\|\frac{\partial y_k^t}{\partial x_k} - \frac{\partial y^*(x_k)}{\partial x_k}\right\| \overset{(i)}{\leq} \left\|I - \alpha \nabla_y^2 g(x_k, y_k^{t-1})\right\|\left\|\frac{\partial y_k^{t-1}}{\partial x_k} - \frac{\partial y^*(x_k)}{\partial x_k}\right\|$$

$$+ \alpha\left(\tau + \frac{L\rho}{\mu}\right)\|y_k^{t-1} - y^*(x_k)\|$$

$$\overset{(ii)}{\leq} (1 - \alpha\mu)\left\|\frac{\partial y_k^{t-1}}{\partial x_k} - \frac{\partial y^*(x_k)}{\partial x_k}\right\| + \alpha\left(\tau + \frac{L\rho}{\mu}\right)\|y_k^{t-1} - y^*(x_k)\|, \tag{39}$$

where $(i)$ follows from Assumption 3 and $(ii)$ follows from the strong-convexity of $g(x, \cdot)$. Based on the strong-convexity of the lower-level function $g(x, \cdot)$, we have

$$\|y_k^{t-1} - y^*(x_k)\| \leq (1 - \alpha\mu)^{\frac{t-1}{2}}\|y_k^0 - y^*(x_k)\|. \tag{40}$$

Substituting eq. (40) into eq. (39) and telecoping eq. (39) over $t$ from 1 to $T$, we have

$$\left\|\frac{\partial y_k^T}{\partial x_k} - \frac{\partial y^*(x_k)}{\partial x_k}\right\| \leq (1 - \alpha\mu)^T\left\|\frac{\partial y_k^0}{\partial x_k} - \frac{\partial y^*(x_k)}{\partial x_k}\right\|$$

$$+ \alpha\left(\tau + \frac{L\rho}{\mu}\right)\sum_{t=0}^{T-1}(1 - \alpha\mu)^{T-1-t}(1 - \alpha\mu)^{\frac{t}{2}}\|y_k^0 - y^*(x_k)\|$$

$$= (1 - \alpha\mu)^T\left\|\frac{\partial y_k^0}{\partial x_k} - \frac{\partial y^*(x_k)}{\partial x_k}\right\| + \frac{2(\tau\mu + L\rho)}{\mu^2}(1 - \alpha\mu)^{\frac{T-1}{2}}\|y_k^0 - y^*(x_k)\|$$

$$\leq \frac{L(1 - \alpha\mu)^T}{\mu} + \frac{2(\tau\mu + L\rho)}{\mu^2}(1 - \alpha\mu)^{\frac{T-1}{2}}\|y_k^0 - y^*(x_k)\|, \tag{41}$$

where the last inequality follows from $\frac{\partial y_k^0}{\partial x_k} = 0$ and eq. (38). Then, combining eq. (34), eq. (38), eq. (40) and eq. (41) completes the proof. $\square$

### F.1 PROOF OF THEOREM 2

Based on the characterization on the estimation error of the gradient estimate $\frac{\partial f(x_k, y_k^T)}{\partial x_k}$ in Lemma 6, we now prove Theorem 2.

Recall the notation that $\widehat{\nabla}\Phi(x_k) = \frac{\partial f(x_k, y_k^T)}{\partial x_k}$. Using an approach similar to eq. (28), we have

$$\Phi(x_{k+1}) \leq \Phi(x_k) - \left(\frac{\beta}{2} - \beta^2 L_\Phi\right)\|\nabla\Phi(x_k)\|^2 + \left(\frac{\beta}{2} + \beta^2 L_\Phi\right)\|\nabla\Phi(x_k) - \widehat{\nabla}\Phi(x_k)\|^2, \tag{42}$$

which, in conjunction with Lemma 6 and use $\|y_k^0 - y^*(x_k)\|^2 \leq \Delta$, yields

$$\Phi(x_{k+1}) \leq \Phi(x_k) - \left(\frac{\beta}{2} - \beta^2 L_\Phi\right)\|\nabla\Phi(x_k)\|^2$$

$$+ 3\Delta\left(\frac{\beta}{2} + \beta^2 L_\Phi\right)\left(\frac{L^2(L+\mu)^2}{\mu^2}(1 - \alpha\mu)^T + \frac{4M^2(\tau\mu + L\rho)^2}{\mu^4}(1 - \alpha\mu)^{T-1}\right)$$

$$+ 3\left(\frac{\beta}{2} + \beta^2 L_\Phi\right)\frac{L^2 M^2(1 - \alpha\mu)^{2T}}{\mu^2}. \tag{43}$$

Telescoping eq. (43) over $k$ from 0 to $K - 1$ yields

$$\frac{1}{K}\sum_{k=0}^{K-1}\left(\frac{1}{2} - \beta L_\Phi\right)\|\nabla\Phi(x_k)\|^2 \leq \frac{\Phi(x_0) - \inf_x \Phi(x)}{\beta K} + 3\left(\frac{1}{2} + \beta L_\Phi\right)\frac{L^2 M^2(1 - \alpha\mu)^{2T}}{\mu^2}$$

$$+ 3\Delta\left(\frac{1}{2} + \beta L_\Phi\right)\left(\frac{L^2(L+\mu)^2}{\mu^2}(1 - \alpha\mu)^T + \frac{4M^2(\tau\mu + L\rho)^2}{\mu^4}(1 - \alpha\mu)^{T-1}\right). \tag{44}$$

Substuting $\beta = \frac{1}{4L_\Phi}$ and $T = \log\left(\max\left\{\frac{3LM}{\mu}, 9\Delta L^2(1+\frac{L}{\mu})^2, \frac{36\Delta M^2(\tau\mu+L\rho)^2}{(1-\alpha\mu)\mu^4}\right\}\frac{9}{2\epsilon}\right)/\log\frac{1}{1-\alpha\mu} = \Theta(\kappa\log\frac{1}{\epsilon})$ in eq. (44) yields

$$\frac{1}{K}\sum_{k=0}^{K-1}\|\nabla\Phi(x_k)\|^2 \le \frac{16L_\Phi(\Phi(x_0) - \inf_x \Phi(x))}{K} + \frac{2\epsilon}{3}. \tag{45}$$

In order to achieve an $\epsilon$-accurate stationary point, we obtain from eq. (45) that ITD-BiO requires at most the total number $K = \mathcal{O}(\kappa^3\epsilon^{-1})$ of outer iterations. Then, based on the gradient form given by Proposition 2, we have the following complexity results.

- Gradient complexity: $\text{Gc}(f, \epsilon) = 2K = \mathcal{O}(\kappa^3\epsilon^{-1})$, $\text{Gc}(g, \epsilon) = KT = \mathcal{O}\left(\kappa^4\epsilon^{-1}\log\frac{1}{\epsilon}\right)$.

- Jacobian- and Hessian-vector product complexities:

$$\text{JV}(g, \epsilon) = KT = \mathcal{O}\left(\kappa^4\epsilon^{-1}\log\frac{1}{\epsilon}\right), \text{HV}(g, \epsilon) = KT = \mathcal{O}\left(\kappa^4\epsilon^{-1}\log\frac{1}{\epsilon}\right).$$

Then, the proof is complete.

# G   PROOFS OF MAIN RESULTS FOR STOCHASTIC CASE IN SECTION 4.2

In this section, we provide proofs for the convergence and complexity results of the proposed algorithm for the stochastic case.

## G.1   PROOF OF PROPOSITION 3

Based on the definition of $v_Q$ in eq. (5) and conditioning on $x_k, y_k^T$, we have

$$\begin{aligned}
\mathbb{E}v_Q &= \mathbb{E}\eta\sum_{q=-1}^{Q-1}\prod_{j=Q-q}^{Q}(I - \eta\nabla_y^2 G(x_k, y_k^T; \mathcal{B}_j))\nabla_y F(x_k, y_k^T; \mathcal{D}_F),\\
&= \eta\sum_{q=0}^{Q}(I - \eta\nabla_y^2 g(x_k, y_k^T))^q\nabla_y f(x_k, y_k^T)\\
&= \eta\sum_{q=0}^{\infty}(I - \eta\nabla_y^2 g(x_k, y_k^T))^q\nabla_y f(x_k, y_k^T) - \eta\sum_{q=Q+1}^{\infty}(I - \eta\nabla_y^2 g(x_k, y_k^T))^q\nabla_y f(x_k, y_k^T)\\
&= \eta(\eta\nabla_y^2 g(x_k, y_k^T))^{-1}\nabla_y f(x_k, y_k^T) - \eta\sum_{q=Q+1}^{\infty}(I - \eta\nabla_y^2 g(x_k, y_k^T))^q\nabla_y f(x_k, y_k^T),
\end{aligned}$$

which, in conjunction with the strong-convexity of function $g(x, \cdot)$, yields

$$\left\|\mathbb{E}v_Q - [\nabla_y^2 g(x_k, y_k^T)]^{-1}\nabla_y f(x_k, y_k^T)\right\| \le \eta\sum_{q=Q+1}^{\infty}(1 - \eta\mu)^q M \le \frac{(1 - \eta\mu)^{Q+1}M}{\mu}. \tag{46}$$

This finishes the proof for the estimation bias. We next prove the variance bound. Note that

$$
\mathbb{E}\left\|\eta\sum_{q=-1}^{Q-1}\prod_{j=Q-q}^{Q}(I-\eta\nabla_y^2 G(x_k,y_k^T;\mathcal{B}_j))\nabla_y F(x_k,y_k^T;\mathcal{D}_F)-(\nabla_y^2 g(x_k,y_k^T))^{-1}\nabla_y f(x_k,y_k^T)\right\|^2
$$

$$
\overset{(i)}{\leq}2\mathbb{E}\left\|\eta\sum_{q=-1}^{Q-1}\prod_{j=Q-q}^{Q}(I-\eta\nabla_y^2 G(x_k,y_k^T;\mathcal{B}_j))-(\nabla_y^2 g(x_k,y_k^T))^{-1}\right\|^2 M^2+\frac{2M^2}{\mu^2 D_f}
$$

$$
\leq 4\mathbb{E}\left\|\eta\sum_{q=-1}^{Q-1}\prod_{j=Q-q}^{Q}(I-\eta\nabla_y^2 G(x_k,y_k^T;\mathcal{B}_j))-\eta\sum_{q=0}^{Q}(I-\eta\nabla_y^2 g(x_k,y_k^T))^q\right\|^2 M^2
$$

$$
+4\mathbb{E}\left\|\eta\prod_{q=0}^{Q}(I-\eta\nabla_y^2 g(x_k,y_k^T))^q)-(\nabla_y^2 g(x_k,y_k^T))^{-1}\right\|^2 M^2+\frac{2M^2}{\mu^2 D_f}
$$

$$
\overset{(ii)}{\leq}4\eta^2\mathbb{E}\left\|\sum_{q=0}^{Q}\prod_{j=Q+1-q}^{Q}(I-\eta\nabla_y^2 G(x_k,y_k^T;\mathcal{B}_j))-\sum_{q=0}^{Q}(I-\eta\nabla_y^2 g(x_k,y_k^T))^q\right\|^2 M^2+\frac{4(1-\eta\mu)^{2Q+2}M^2}{\mu^2}+\frac{2M^2}{\mu^2 D_f}
$$

$$
\overset{(iii)}{\leq}4\eta^2 M^2 Q\mathbb{E}\sum_{q=0}^{Q}\underbrace{\left\|\prod_{j=Q+1-q}^{Q}(I-\eta\nabla_y^2 G(x_k,y_k^T;\mathcal{B}_j))-(I-\eta\nabla_y^2 g(x_k,y_k^T))^q\right\|^2}_{M_q}
$$

$$
+\frac{4(1-\eta\mu)^{2Q+2}M^2}{\mu^2}+\frac{2M^2}{\mu^2 D_f} \tag{47}
$$

where $(i)$ follows from Lemma 1, $(ii)$ follows from eq. (46), and $(iii)$ follows from the Cauchy-Schwarz inequality.

Our next step is to upper-bound $M_q$ in eq. (47). For simplicity, we define a general quantity $M_i$ for by replacing $q$ in $M_q$ with $i$. Then, we have

$$
\mathbb{E}M_i=\mathbb{E}\left\|(I-\eta\nabla_y^2 g(x_k,y_k^T))\prod_{j=Q+2-i}^{Q}(I-\eta\nabla_y^2 G(x_k,y_k^T;\mathcal{B}_j))-(I-\eta\nabla_y^2 g(x_k,y_k^T))^i\right\|^2
$$

$$
+\mathbb{E}\left\|\eta(\nabla_y^2 g(x_k,y_k^T)-\nabla_y^2 G(x_k,y_k^T;\mathcal{B}_{Q+1-i}))\prod_{j=Q+2-i}^{Q}(I-\eta\nabla_y^2 G(x_k,y_k^T;\mathcal{B}_j))\right\|^2
$$

$$
+2\mathbb{E}\Big\langle(I-\eta\nabla_y^2 g(x_k,y_k^T))\prod_{j=Q+2-i}^{Q}(I-\eta\nabla_y^2 G(x_k,y_k^T;\mathcal{B}_j))-(I-\eta\nabla_y^2 g(x_k,y_k^T))^i,
$$

$$
\eta(\nabla_y^2 g(x_k,y_k^T)-\nabla_y^2 G(x_k,y_k^T;\mathcal{B}_{Q+1-i}))\prod_{j=Q+2-i}^{Q}(I-\eta\nabla_y^2 G(x_k,y_k^T;\mathcal{B}_j))\Big\rangle
$$

$$
\overset{(i)}{=}\mathbb{E}\left\|(I-\eta\nabla_y^2 g(x_k,y_k^T))\prod_{j=Q+2-i}^{Q}(I-\eta\nabla_y^2 G(x_k,y_k^T;\mathcal{B}_j))-(I-\eta\nabla_y^2 g(x_k,y_k^T))^i\right\|^2
$$

$$
+\mathbb{E}\left\|\eta(\nabla_y^2 g(x_k,y_k^T)-\nabla_y^2 G(x_k,y_k^T;\mathcal{B}_{Q+1-i}))\prod_{j=Q+2-i}^{Q}(I-\eta\nabla_y^2 G(x_k,y_k^T;\mathcal{B}_j))\right\|^2
$$

$$
\overset{(ii)}{\leq}(1-\eta\mu)^2\mathbb{E}M_{i-1}+\eta^2(1-\eta\mu)^{2i-2}\mathbb{E}\|\nabla_y^2 g(x_k,y_k^T)-\nabla_y^2 G(x_k,y_k^T;\mathcal{B}_{Q+1-i})\|^2
$$

$$
\overset{(iii)}{\leq}(1-\eta\mu)^2\mathbb{E}M_{i-1}+\eta^2(1-\eta\mu)^{2i-2}\frac{L^2}{|\mathcal{B}_{Q+1-i}|}, \tag{48}
$$

where $(i)$ follows from that fact that $\mathbb{E}_{\mathcal{B}_{Q+1-i}}\nabla_y^2 G(x_k,y_k^T;\mathcal{B}_{Q+1-i})=\nabla_y^2 g(x_k,y_k^T)$, $(ii)$ follows from the strong-convexity of function $G(x,\cdot;\xi)$, and $(iii)$ follows from Lemma 1.

Then, telescoping eq. (48) over $i$ from 2 to $q$ yields

$$\mathbb{E}M_q \leq L^2\eta^2(1-\eta\mu)^{2q-2}\sum_{j=1}^{q}\frac{1}{|\mathcal{B}_{Q+1-j}|},$$

which, in conjunction with the choice of $|\mathcal{B}_{Q+1-j}| = BQ(1-\eta\mu)^{j-1}$ for $j = 1, ..., Q$, yields

$$\mathbb{E}M_q \leq \eta^2(1-\eta\mu)^{2q-2}\sum_{j=1}^{q}\frac{L^2}{BQ}\Big(\frac{1}{1-\eta\mu}\Big)^{j-1}$$

$$=\frac{\eta^2 L^2}{BQ}(1-\eta\mu)^{2q-2}\frac{\Big(\frac{1}{1-\eta\mu}\Big)^{q-1}-1}{\frac{1}{1-\eta\mu}-1} \leq \frac{\eta L^2}{(1-\eta\mu)\mu}\frac{1}{BQ}(1-\eta\mu)^q. \tag{49}$$

Substituting eq. (49) into eq. (47) yields

$$\mathbb{E}\Big\|\eta\sum_{q=-1}^{Q-1}\prod_{j=Q-q}^{Q}(I-\eta\nabla_y^2 G(x_k, y_k^T; \mathcal{B}_j))\nabla_y F(x_k, y_k^T; \mathcal{D}_F) - (\nabla_y^2 g(x_k, y_k^T))^{-1}\nabla_y f(x_k, y_k^T)\Big\|^2$$

$$\leq 4\eta^2 M^2 Q\sum_{q=0}^{Q}\frac{\eta L^2}{(1-\eta\mu)\mu}\frac{1}{BQ}(1-\eta\mu)^q + \frac{4(1-\eta\mu)^{2Q+2}M^2}{\mu^2} + \frac{2M^2}{\mu^2 D_f}$$

$$\leq \frac{4\eta^2 L^2 M^2}{\mu^2}\frac{1}{B} + \frac{4(1-\eta\mu)^{2Q+2}M^2}{\mu^2} + \frac{2M^2}{\mu^2 D_f}, \tag{50}$$

where the last inequality follows from the fact that $\sum_{q=0}^{S}x^q \leq \frac{1}{1-x}$. Then, the proof is complete.

## G.2 AUXILIARY LEMMAS FOR PROVING THEOREM 3

We first use the following lemma to characterize the first-moment error of the gradient estimate $\widehat{\nabla}\Phi(x_k)$, whose form is given by eq. (6).

**Lemma 7.** *Suppose Assumptions 1, 2 and 3 hold. Then, conditioning on $x_k$ and $y_k^T$, we have*

$$\big\|\mathbb{E}\widehat{\nabla}\Phi(x_k) - \nabla\Phi(x_k)\big\|^2 \leq 2\Big(L + \frac{L^2}{\mu} + \frac{M\tau}{\mu} + \frac{LM\rho}{\mu^2}\Big)^2\|y_k^T - y^*(x_k)\|^2 + \frac{2L^2 M^2(1-\eta\mu)^{2Q}}{\mu^2}.$$

**Proof of Lemma 7.** To simplify notations, we define

$$\widetilde{\nabla}\Phi_T(x_k) = \nabla_x f(x_k, y_k^T) - \nabla_x\nabla_y g(x_k, y_k^T)\big[\nabla_y^2 g(x_k, y_k^T)\big]^{-1}\nabla_y f(x_k, y_k^T). \tag{51}$$

Based on the definition of $\widehat{\nabla}\Phi(x_k)$ in eq. (6) and conditioning on $x_k$ and $y_k^T$, we have

$$\mathbb{E}\widehat{\nabla}\Phi(x_k) = \nabla_x f(x_k, y_k^T) - \nabla_x\nabla_y g(x_k, y_k^T)\mathbb{E}v_Q$$

$$= \widetilde{\nabla}\Phi_T(x_k) - \nabla_x\nabla_y g(x_k, y_k^T)(\mathbb{E}v_Q - [\nabla_y^2 g(x_k, y_k^T)]^{-1}\nabla_y f(x_k, y_k^T)),$$

which further implies that

$$\big\|\mathbb{E}\widehat{\nabla}\Phi(x_k) - \nabla\Phi(x_k)\big\|^2$$

$$\leq 2\mathbb{E}\|\widetilde{\nabla}\Phi_T(x_k) - \nabla\Phi(x_k)\|^2 + 2\|\mathbb{E}\widehat{\nabla}\Phi(x_k) - \widetilde{\nabla}\Phi_T(x_k)\|^2$$

$$\leq 2\mathbb{E}\|\widetilde{\nabla}\Phi_T(x_k) - \nabla\Phi(x_k)\|^2 + 2L^2\|\mathbb{E}v_Q - [\nabla_y^2 g(x_k, y_k^T)]^{-1}\nabla_y f(x_k, y_k^T)\|^2$$

$$\leq 2\mathbb{E}\|\widetilde{\nabla}\Phi_T(x_k) - \nabla\Phi(x_k)\|^2 + \frac{2L^2 M^2(1-\eta\mu)^{2Q+2}}{\mu^2}, \tag{52}$$

where the last inequality follows from Proposition 3. Our next step is to upper-bound the first term at the right hand side of eq. (52). Using the fact that $\big\|\nabla_y^2 g(x, y)^{-1}\big\| \leq \frac{1}{\mu}$ and based on Assumptions 2

and 3, we have

$$
\begin{aligned}
\|\widetilde{\nabla}\Phi_T(x_k) - \nabla\Phi(x_k)\| \leq & \|\nabla_x f(x_k, y_k^T) - \nabla_x f(x_k, y^*(x_k))\| \\
& + \frac{L^2}{\mu}\|y_k^T - y^*(x_k)\| + \frac{M\tau}{\mu}\|y_k^T - y^*(x_k)\| \\
& + LM\|\nabla_y^2 g(x_k, y_k^T)^{-1} - \nabla_y^2 g(x_k, y^*(x_k))^{-1}\| \\
\leq & \Big(L + \frac{L^2}{\mu} + \frac{M\tau}{\mu} + \frac{LM\rho}{\mu^2}\Big)\|y_k^T - y^*(x_k)\|,
\end{aligned}
\tag{53}
$$

where the last inequality follows from the inequality $\|M_1^{-1} - M_2^{-1}\| \leq \|M_1^{-1} M_2^{-1}\|\|M_1 - M_2\|$ for any two matrices $M_1$ and $M_2$. Combining eq. (52) and eq. (53) yields

$$
\big\|\mathbb{E}\widehat{\nabla}\Phi(x_k) - \nabla\Phi(x_k)\big\|^2 \leq 2\Big(L + \frac{L^2}{\mu} + \frac{M\tau}{\mu} + \frac{LM\rho}{\mu^2}\Big)^2\|y_k^T - y^*(x_k)\|^2 + \frac{2L^2 M^2 (1 - \eta\mu)^{2Q}}{\mu^2},
$$

which completes the proof. $\qquad\square$

Then, we use the following lemma to characterize the variance of the estimator $\widehat{\nabla}\Phi(x_k)$.

**Lemma 8.** *Suppose Assumptions 1, 2 and 3 hold. Then, we have*

$$
\begin{aligned}
\mathbb{E}\|\widehat{\nabla}\Phi(x_k) - \nabla\Phi(x_k)\|^2 \leq & \frac{4L^2 M^2}{\mu^2 D_g} + \Big(\frac{8L^2}{\mu^2} + 2\Big)\frac{M^2}{D_f} + \frac{16\eta^2 L^4 M^2}{\mu^2}\frac{1}{B} + \frac{16 L^2 M^2 (1 - \eta\mu)^{2Q}}{\mu^2} \\
& + \Big(L + \frac{L^2}{\mu} + \frac{M\tau}{\mu} + \frac{LM\rho}{\mu^2}\Big)^2 \mathbb{E}\|y_k^T - y^*(x_k)\|^2.
\end{aligned}
$$

**Proof of Lemma 8.** Based on the definitions of $\nabla\Phi(x_k)$ and $\widetilde{\nabla}\Phi_T(x_k)$ in eq. (4) and eq. (51) and conditioning on $x_k$ and $y_k^T$, we have

$$
\begin{aligned}
& \mathbb{E}\|\widehat{\nabla}\Phi(x_k) - \nabla\Phi(x_k)\|^2 \\
& \overset{(i)}{=} \mathbb{E}\|\widehat{\nabla}\Phi(x_k) - \widetilde{\nabla}\Phi_T(x_k)\|^2 + \|\widetilde{\nabla}\Phi_T(x_k) - \nabla\Phi(x_k)\|^2 \\
& \overset{(ii)}{\leq} 2\mathbb{E}\big\|\nabla_x \nabla_y G(x_k, y_k^T; \mathcal{D}_G)v_Q - \nabla_x\nabla_y g(x_k, y_k^T)\big[\nabla_y^2 g(x_k, y_k^T)\big]^{-1}\nabla_y f(x_k, y_k^T)\big\|^2 + \frac{2M^2}{D_f} \\
& \quad + \Big(L + \frac{L^2}{\mu} + \frac{M\tau}{\mu} + \frac{LM\rho}{\mu^2}\Big)^2\|y_k^T - y^*(x_k)\|^2 \\
& \overset{(iii)}{\leq} \frac{4M^2}{\mu^2}\mathbb{E}\|\nabla_x\nabla_y G(x_k, y_k^T; \mathcal{D}_G) - \nabla_x\nabla_y g(x_k, y_k^T)\|^2 + 4L^2\mathbb{E}\|v_Q - \big[\nabla_y^2 g(x_k, y_k^T)\big]^{-1}\nabla_y f(x_k, y_k^T)\|^2 \\
& \quad + \Big(L + \frac{L^2}{\mu} + \frac{M\tau}{\mu} + \frac{LM\rho}{\mu^2}\Big)^2\|y_k^T - y^*(x_k)\|^2 + \frac{2M^2}{D_f},
\end{aligned}
\tag{54}
$$

where $(i)$ follows from the fact that $\mathbb{E}_{\mathcal{D}_G, \mathcal{D}_H, \mathcal{D}_F}\widehat{\nabla}\Phi(x_k) = \widetilde{\nabla}\Phi_T(x_k)$, $(ii)$ follows from Lemma 1 and eq. (53), and $(iii)$ follows from the Young's inequality and Assumption 2.

Using Lemma 1 and Proposition 3 in eq. (54), yields

$$
\begin{aligned}
\mathbb{E}\|\widehat{\nabla}\Phi(x_k) - \nabla\Phi(x_k)\|^2 \leq & \frac{4L^2 M^2}{\mu^2 D_g} + \frac{16\eta^2 L^4 M^2}{\mu^2}\frac{1}{B} + \frac{16(1 - \eta\mu)^{2Q} L^2 M^2}{\mu^2} + \frac{8L^2 M^2}{\mu^2 D_f} \\
& + \Big(L + \frac{L^2}{\mu} + \frac{M\tau}{\mu} + \frac{LM\rho}{\mu^2}\Big)^2\|y_k^T - y^*(x_k)\|^2 + \frac{2M^2}{D_f},
\end{aligned}
\tag{55}
$$

which, unconditioning on $x_k$ and $y_k^T$, completes the proof. $\qquad\square$

It can be seen from Lemmas 7 and 8 that the upper bounds on both the estimation error and bias depend on the tracking error $\|y_k^T - y^*(x_k)\|^2$. The following lemma provides an upper bound on such tracking error $\|y_k^T - y^*(x_k)\|^2$.

**Lemma 9.** *Suppose Assumptions 1, 2 and 4 hold. Define constants*

$$\lambda = \left(\frac{L-\mu}{L+\mu}\right)^{2T}\left(2 + \frac{4\beta^2 L^2}{\mu^2}\left(L + \frac{L^2}{\mu} + \frac{M\tau}{\mu} + \frac{LM\rho}{\mu^2}\right)^2\right)$$

$$\Delta = \frac{4L^2 M^2}{\mu^2 D_g} + \left(\frac{8L^2}{\mu^2} + 2\right)\frac{M^2}{D_f} + \frac{16\eta^2 L^4 M^2}{\mu^2}\frac{1}{B} + \frac{16L^2 M^2(1-\eta\mu)^{2Q}}{\mu^2}$$

$$\omega = \frac{4\beta^2 L^2}{\mu^2}\left(\frac{L-\mu}{L+\mu}\right)^{2T}. \tag{56}$$

*Choose $T$ such that $\lambda < 1$ and set inner-loop stepsize $\alpha = \frac{2}{L+\mu}$. Then, we have*

$$\mathbb{E}\|y_k^T - y^*(x_k)\|^2$$

$$\leq \lambda^k \left(\left(\frac{L-\mu}{L+\mu}\right)^{2T}\|y_0 - y^*(x_0)\|^2 + \frac{\sigma^2}{L\mu S}\right) + \omega\sum_{j=0}^{k-1}\lambda^{k-1-j}\mathbb{E}\|\nabla\Phi(x_j)\|^2 + \frac{\omega\Delta + \frac{\sigma^2}{L\mu S}}{1-\lambda}.$$

**Proof of Lemma 9.** First note that for an integer $t \leq T$

$$\|y_k^{t+1} - y^*(x_k)\|^2$$

$$= \|y_k^{t+1} - y_k^t\|^2 + 2\langle y_k^{t+1} - y_k^t, y_k^t - y^*(x_k)\rangle + \|y_k^t - y^*(x_k)\|^2$$

$$= \alpha^2\|\nabla_y G(x_k, y_k^t; \mathcal{S}_t)\|^2 - 2\alpha\langle\nabla_y G(x_k, y_k^t; \mathcal{S}_t), y_k^t - y^*(x_k)\rangle + \|y_k^t - y^*(x_k)\|^2. \tag{57}$$

Conditioning on $y_k^t$ and taking expectation in eq. (57), we have

$$\mathbb{E}\|y_k^{t+1} - y^*(x_k)\|^2$$

$$\overset{(i)}{\leq} \alpha^2\left(\frac{\sigma^2}{S} + \|\nabla_y g(x_k, y_k^t)\|^2\right) - 2\alpha\langle\nabla_y g(x_k, y_k^t), y_k^t - y^*(x_k)\rangle$$

$$\quad + \|y_k^t - y^*(x_k)\|^2$$

$$\overset{(ii)}{\leq} \frac{\alpha^2\sigma^2}{S} + \alpha^2\|\nabla_y g(x_k, y_k^t)\|^2 - 2\alpha\left(\frac{L\mu}{L+\mu}\|y_k^t - y^*(x_k)\|^2 + \frac{\|\nabla_y g(x_k, y_k^t)\|^2}{L+\mu}\right)$$

$$\quad + \|y_k^t - y^*(x_k)\|^2$$

$$= \frac{\alpha^2\sigma^2}{S} - \alpha\left(\frac{2}{L+\mu} - \alpha\right)\|\nabla_y g(x_k, y_k^t)\|^2 + \left(1 - \frac{2\alpha L\mu}{L+\mu}\right)\|y_k^t - y^*(x_k)\|^2 \tag{58}$$

where $(i)$ follows from the third item in Assumption 2, $(ii)$ follows from the strong-convexity and smoothness of the function $g$. Since $\alpha = \frac{2}{L+\mu}$, we obtain from eq. (58) that

$$\mathbb{E}\|y_k^{t+1} - y^*(x_k)\|^2 \leq \left(\frac{L-\mu}{L+\mu}\right)^2\|y_k^t - y^*(x_k)\|^2 + \frac{4\sigma^2}{(L+\mu)^2 S}. \tag{59}$$

Unconditioning on $y_k^t$ in eq. (59) and telescoping eq. (59) over $t$ from 0 to $T-1$ yields

$$\mathbb{E}\|y_k^T - y^*(x_k)\|^2 \leq \left(\frac{L-\mu}{L+\mu}\right)^{2T}\mathbb{E}\|y_k^0 - y^*(x_k)\|^2 + \frac{\sigma^2}{L\mu S}$$

$$= \left(\frac{L-\mu}{L+\mu}\right)^{2T}\mathbb{E}\|y_{k-1}^T - y^*(x_k)\|^2 + \frac{\sigma^2}{L\mu S}, \tag{60}$$

where the last inequality follows from Algorithm 2 that $y_k^0 = y_{k-1}^T$. Note that

$$\mathbb{E}\|y_{k-1}^T - y^*(x_k)\|^2 \leq 2\mathbb{E}\|y_{k-1}^T - y^*(x_{k-1})\|^2 + 2\mathbb{E}\|y^*(x_{k-1}) - y^*(x_k)\|^2$$

$$\overset{(i)}{\leq} 2\mathbb{E}\|y_{k-1}^T - y^*(x_{k-1})\|^2 + \frac{2L^2}{\mu^2}\mathbb{E}\|x_k - x_{k-1}\|^2$$

$$\leq 2\mathbb{E}\|y_{k-1}^T - y^*(x_{k-1})\|^2 + \frac{2\beta^2 L^2}{\mu^2}\mathbb{E}\|\widehat{\nabla}\Phi(x_{k-1})\|^2$$

$$\leq 2\mathbb{E}\|y_{k-1}^T - y^*(x_{k-1})\|^2 + \frac{4\beta^2 L^2}{\mu^2}\mathbb{E}\|\nabla\Phi(x_{k-1})\|^2$$

$$\quad + \frac{4\beta^2 L^2}{\mu^2}\mathbb{E}\|\widehat{\nabla}\Phi(x_{k-1}) - \nabla\Phi(x_{k-1})\|^2, \tag{61}$$

where $(i)$ follows from Lemma 2.2 in Ghadimi & Wang (2018). Using Lemma 8 in eq. (61) yields

$$\mathbb{E}\|y_{k-1}^T - y^*(x_k)\|^2$$

$$\leq \left(2 + \frac{4\beta^2 L^2}{\mu^2}\left(L + \frac{L^2}{\mu} + \frac{M\tau}{\mu} + \frac{LM\rho}{\mu^2}\right)^2\right)\mathbb{E}\|y_{k-1}^T - y^*(x_{k-1})\|^2 + \frac{4\beta^2 L^2}{\mu^2}\mathbb{E}\|\nabla\Phi(x_{k-1})\|^2$$

$$+ \frac{4\beta^2 L^2}{\mu^2}\left(\frac{4L^2M^2}{\mu^2 D_g} + \left(\frac{8L^2}{\mu^2} + 2\right)\frac{M^2}{D_f} + \frac{16\eta^2 L^4 M^2}{\mu^2}\frac{1}{B} + \frac{16L^2 M^2(1-\eta\mu)^{2Q}}{\mu^2}\right). \quad (62)$$

Combining eq. (60) and eq. (62) yields

$$\mathbb{E}\|y_k^T - y^*(x_k)\|^2$$

$$\leq \left(\frac{L-\mu}{L+\mu}\right)^{2T}\left(2 + \frac{4\beta^2 L^2}{\mu^2}\left(L + \frac{L^2}{\mu} + \frac{M\tau}{\mu} + \frac{LM\rho}{\mu^2}\right)^2\right)\mathbb{E}\|y_{k-1}^T - y^*(x_{k-1})\|^2$$

$$+ \left(\frac{L-\mu}{L+\mu}\right)^{2T}\frac{4\beta^2 L^2}{\mu^2}\left(\frac{4L^2M^2}{\mu^2 D_g} + \left(\frac{8L^2}{\mu^2} + 2\right)\frac{M^2}{D_f} + \frac{16\eta^2 L^4 M^2}{\mu^2}\frac{1}{B} + \frac{16L^2 M^2(1-\eta\mu)^{2Q}}{\mu^2}\right)$$

$$+ \frac{4\beta^2 L^2}{\mu^2}\left(\frac{L-\mu}{L+\mu}\right)^{2T}\mathbb{E}\|\nabla\Phi(x_{k-1})\|^2 + \frac{\sigma^2}{L\mu S}. \quad (63)$$

Based on the definitions of $\lambda, \omega, \Delta$ in eq. (56), we obtain from eq. (63) that

$$\mathbb{E}\|y_k^T - y^*(x_k)\|^2 \leq \lambda\mathbb{E}\|y_{k-1}^T - y^*(x_{k-1})\|^2 + \omega\Delta + \frac{\sigma^2}{L\mu S} + \omega\mathbb{E}\|\nabla\Phi(x_{k-1})\|^2. \quad (64)$$

Telescoping eq. (64) over $k$ yields

$$\mathbb{E}\|y_k^T - y^*(x_k)\|^2$$

$$\leq \lambda^k\mathbb{E}\|y_0^T - y^*(x_0)\|^2 + \omega\sum_{j=0}^{k-1}\lambda^{k-1-j}\mathbb{E}\|\nabla\Phi(x_j)\|^2 + \frac{\omega\Delta + \frac{\sigma^2}{L\mu S}}{1-\lambda}$$

$$\leq \lambda^k\left(\left(\frac{L-\mu}{L+\mu}\right)^{2T}\|y_0 - y^*(x_0)\|^2 + \frac{\sigma^2}{L\mu S}\right) + \omega\sum_{j=0}^{k-1}\lambda^{k-1-j}\mathbb{E}\|\nabla\Phi(x_j)\|^2 + \frac{\omega\Delta + \frac{\sigma^2}{L\mu S}}{1-\lambda},$$

which completes the proof. $\qquad\square$

### G.3 Proof of Theorem 3

In this subsection, we provide the proof for Theorem 3, based on the supporting lemmas we develop in Appendix G.2.

Based on the smoothness of the function $\Phi(x)$ in Lemma 2, we have

$$\Phi(x_{k+1}) \leq \Phi(x_k) + \langle\nabla\Phi(x_k), x_{k+1} - x_k\rangle + \frac{L_\Phi}{2}\|x_{k+1} - x_k\|^2$$

$$\leq \Phi(x_k) - \beta\langle\nabla\Phi(x_k), \widehat{\nabla}\Phi(x_k)\rangle + \beta^2 L_\Phi\|\nabla\Phi(x_k)\|^2 + \beta^2 L_\Phi\|\nabla\Phi(x_k) - \widehat{\nabla}\Phi(x_k)\|^2.$$

For simplicity, let $\mathbb{E}_k = \mathbb{E}(\cdot\,|\,x_k, y_k^T)$. Note that we choose $\beta = \frac{1}{4L_\phi}$. Then, taking expectation over the above inequality, we have

$$\mathbb{E}\Phi(x_{k+1}) \leq \mathbb{E}\Phi(x_k) - \beta\mathbb{E}\langle\nabla\Phi(x_k), \mathbb{E}_k\widehat{\nabla}\Phi(x_k)\rangle + \beta^2 L_\Phi\mathbb{E}\|\nabla\Phi(x_k)\|^2$$

$$+ \beta^2 L_\Phi\mathbb{E}\|\nabla\Phi(x_k) - \widehat{\nabla}\Phi(x_k)\|^2$$

$$\overset{(i)}{\leq} \mathbb{E}\Phi(x_k) + \frac{\beta}{2}\mathbb{E}\|\mathbb{E}_k\widehat{\nabla}\Phi(x_k) - \nabla\Phi(x_k)\|^2 - \frac{\beta}{4}\mathbb{E}\|\nabla\Phi(x_k)\|^2 + \frac{\beta}{4}\mathbb{E}\|\nabla\Phi(x_k) - \widehat{\nabla}\Phi(x_k)\|^2$$

$$\overset{(ii)}{\leq} \mathbb{E}\Phi(x_k) - \frac{\beta}{4}\mathbb{E}\|\nabla\Phi(x_k)\|^2 + \frac{\beta L^2 M^2(1-\eta\mu)^{2Q}}{\mu^2}$$

$$+ \frac{\beta}{4}\left(\frac{4L^2M^2}{\mu^2 D_g} + \left(\frac{8L^2}{\mu^2} + 2\right)\frac{M^2}{D_f} + \frac{16\eta^2 L^4 M^2}{\mu^2}\frac{1}{B} + \frac{16L^2 M^2(1-\eta\mu)^{2Q}}{\mu^2}\right)$$

$$+ \frac{5\beta}{4}\left(L + \frac{L^2}{\mu} + \frac{M\tau}{\mu} + \frac{LM\rho}{\mu^2}\right)^2\mathbb{E}\|y_k^T - y^*(x_k)\|^2 \quad (65)$$

where $(i)$ follows from Cauchy-Schwarz inequality, and $(ii)$ follows from Lemma 7 and Lemma 8. To simplify notations, Let

$$\nu = \frac{5}{4}\left(L + \frac{L^2}{\mu} + \frac{M\tau}{\mu} + \frac{LM\rho}{\mu^2}\right)^2. \tag{66}$$

Then, applying Lemma 9 in eq. (65) and using the definitions of $\omega, \Delta, \lambda$ in eq. (56), we have

$$\begin{aligned}
\mathbb{E}\Phi(x_{k+1}) \leq &\mathbb{E}\Phi(x_k) - \frac{\beta}{4}\mathbb{E}\|\nabla\Phi(x_k)\|^2 + \frac{\beta L^2 M^2(1-\eta\mu)^{2Q}}{\mu^2} \\
&+ \frac{\beta}{4}\Delta + \beta\nu\lambda^k\left(\left(\frac{L-\mu}{L+\mu}\right)^{2T}\|y_0 - y^*(x_0)\|^2 + \frac{\sigma^2}{L\mu S}\right) \\
&+ \beta\nu\omega\sum_{j=0}^{k-1}\lambda^{k-1-j}\mathbb{E}\|\nabla\Phi(x_j)\|^2 + \frac{\beta\nu(\omega\Delta + \frac{\sigma^2}{L\mu S})}{1-\lambda},
\end{aligned}$$

Telescoping the above inequality over $k$ from $0$ to $K - 1$ yields

$$\begin{aligned}
\mathbb{E}\Phi(x_K) \leq \Phi(x_0) &- \frac{\beta}{4}\sum_{k=0}^{K-1}\mathbb{E}\|\nabla\Phi(x_k)\|^2 + \beta\nu\omega\sum_{k=1}^{K-1}\sum_{j=0}^{k-1}\lambda^{k-1-j}\mathbb{E}\|\nabla\Phi(x_j)\|^2 \\
&+ \frac{K\beta\Delta}{4} + \left(\left(\frac{L-\mu}{L+\mu}\right)^{2T}\|y_0 - y^*(x_0)\|^2 + \frac{\sigma^2}{L\mu S}\right)\frac{\beta\nu}{1-\lambda} \\
&+ \frac{K\beta L^2 M^2(1-\eta\mu)^{2Q}}{\mu^2} + \frac{K\beta\nu(\omega\Delta + \frac{\sigma^2}{L\mu S})}{1-\lambda},
\end{aligned}$$

which, using the fact that

$$\sum_{k=1}^{K-1}\sum_{j=0}^{k-1}\lambda^{k-1-j}\mathbb{E}\|\nabla\Phi(x_j)\|^2 \leq \left(\sum_{k=0}^{K-1}\lambda^k\right)\sum_{k=0}^{K-1}\mathbb{E}\|\nabla\Phi(x_k)\|^2 < \frac{1}{1-\lambda}\sum_{k=0}^{K-1}\mathbb{E}\|\nabla\Phi(x_k)\|^2,$$

yields

$$\begin{aligned}
\left(\frac{1}{4} - \frac{\nu\omega}{1-\lambda}\right)&\frac{1}{K}\sum_{k=0}^{K-1}\mathbb{E}\|\nabla\Phi(x_k)\|^2 \\
&\leq \frac{\Phi(x_0) - \inf_x \Phi(x)}{\beta K} + \frac{\nu\left(\left(\frac{L-\mu}{L+\mu}\right)^{2T}\|y_0 - y^*(x_0)\|^2 + \frac{\sigma^2}{L\mu S}\right)}{K(1-\lambda)} + \frac{\Delta}{4} + \frac{L^2 M^2(1-\eta\mu)^{2Q}}{\mu^2} \\
&\quad + \frac{\nu(\omega\Delta + \frac{\sigma^2}{L\mu S})}{1-\lambda}.
\end{aligned} \tag{67}$$

Since $\beta = \frac{1}{4L_\Phi}$ and $T \geq \frac{\log\left(12 + \frac{48\beta^2 L^2}{\mu^2}(L + \frac{L^2}{\mu} + \frac{M\tau}{\mu} + \frac{LM\rho}{\mu^2})^2\right)}{2\log(\frac{L+\mu}{L-\mu})}$, we have $\lambda \leq \frac{1}{6}$, and hence eq. (67) is further simplified to

$$\begin{aligned}
\left(\frac{1}{4} - \frac{6}{5}\nu\omega\right)&\frac{1}{K}\sum_{k=0}^{K-1}\mathbb{E}\|\nabla\Phi(x_k)\|^2 \\
&\leq \frac{\Phi(x_0) - \inf_x \Phi(x)}{\beta K} + \frac{2\nu\left(\left(\frac{L-\mu}{L+\mu}\right)^{2T}\|y_0 - y^*(x_0)\|^2 + \frac{\sigma^2}{L\mu S}\right)}{K} + \frac{\Delta}{4} + \frac{L^2 M^2(1-\eta\mu)^{2Q}}{\mu^2} \\
&\quad + 2\nu\left(\omega\Delta + \frac{\sigma^2}{L\mu S}\right).
\end{aligned} \tag{68}$$

By the definitions of $\omega$ in eq. (56) and $\nu$ in eq. (66) and $T \geq \frac{\log\left(12 + \frac{48\beta^2 L^2}{\mu^2}(L + \frac{L^2}{\mu} + \frac{M\tau}{\mu} + \frac{LM\rho}{\mu^2})^2\right)}{2\log(\frac{L+\mu}{L-\mu})}$, we have

$$\nu\omega = \frac{5\beta^2 L^2}{\mu^2}\left(\frac{L-\mu}{L+\mu}\right)^{2T}\left(L + \frac{L^2}{\mu} + \frac{M\tau}{\mu} + \frac{LM\rho}{\mu^2}\right)^2$$

$$< \frac{\frac{5\beta^2 L^2}{\mu^2}\left(L + \frac{L^2}{\mu} + \frac{M\tau}{\mu} + \frac{LM\rho}{\mu^2}\right)^2}{12 + \frac{48\beta^2 L^2}{\mu^2}(L + \frac{L^2}{\mu} + \frac{M\tau}{\mu} + \frac{LM\rho}{\mu^2})^2} \leq \frac{5}{48}. \tag{69}$$

In addition, since $T > \frac{\log\left(\sqrt{\beta}\left(L + \frac{L^2}{\mu} + \frac{M\tau}{\mu} + \frac{LM\rho}{\mu^2}\right)\right)}{\log(\frac{L+\mu}{L-\mu})}$, we have

$$\nu\left(\frac{L-\mu}{L+\mu}\right)^{2T} = \frac{5}{4}\left(\frac{L-\mu}{L+\mu}\right)^{2T}\left(L + \frac{L^2}{\mu} + \frac{M\tau}{\mu} + \frac{LM\rho}{\mu^2}\right)^2 < \frac{5}{4\beta}. \tag{70}$$

Substituting eq. (69) and eq. (70) in eq. (68) yields

$$\frac{1}{K}\sum_{k=0}^{K-1}\mathbb{E}\|\nabla\Phi(x_k)\|^2 \leq \frac{8(\Phi(x_0) - \inf_x \Phi(x) + \frac{5}{2}\|y_0 - y^*(x_0)\|^2)}{\beta K} + \left(1 + \frac{1}{K}\right)\frac{16\nu\sigma^2}{L\mu S}$$

$$+ \frac{11}{3}\Delta + \frac{8L^2 M^2}{\mu^2}(1 - \eta\mu)^{2Q}, \tag{71}$$

which, in conjunction with eq. (56) and eq. (66), yields eq. (10) in Theorem 3.

Then, based on eq. (10), in order to achieve an $\epsilon$-accurate stationary point, i.e., $\mathbb{E}\|\nabla\Phi(\bar{x})\|^2 \leq \epsilon$ with $\bar{x}$ chosen from $x_0, ..., x_{K-1}$ uniformly at random, it suffices to choose

$$K = \frac{32L_\Phi(\Phi(x_0) - \inf_x \Phi(x) + \frac{5}{2}\|y_0 - y^*(x_0)\|^2)}{\epsilon} = \mathcal{O}\left(\frac{\kappa^3}{\epsilon}\right), T = \Theta(\kappa)$$

$$Q = \kappa\log\frac{\kappa^2}{\epsilon}, S = \mathcal{O}\left(\frac{\kappa^5}{\epsilon}\right), D_g = \mathcal{O}\left(\frac{\kappa^2}{\epsilon}\right), D_f = \mathcal{O}\left(\frac{\kappa^2}{\epsilon}\right), B = \mathcal{O}\left(\frac{\kappa^2}{\epsilon}\right).$$

Note that the above choices of $Q$ and $B$ satisfy the condition that $B \geq \frac{1}{Q(1-\eta\mu)^{Q-1}}$ required in Proposition 3.

Then, the gradient complexity is given by $\mathrm{Gc}(F, \epsilon) = KD_f = \mathcal{O}(\kappa^5\epsilon^{-2}), \mathrm{Gc}(G, \epsilon) = KTS = \mathcal{O}(\kappa^9\epsilon^{-2})$. In addition, the Jacobian- and Hessian-vector product complexities are given by $\mathrm{JV}(G, \epsilon) = KD_g = \mathcal{O}(\kappa^5\epsilon^{-2})$ and

$$\mathrm{HV}(G, \epsilon) = K\sum_{j=1}^{Q}BQ(1 - \eta\mu)^{j-1} = \frac{KBQ}{\eta\mu} \leq \mathcal{O}\left(\frac{\kappa^6}{\epsilon^2}\log\frac{\kappa^2}{\epsilon}\right).$$

Then, the proof is complete.

## H PROOF OF THEOREM 4 ON META-LEARNING

To prove Theorem 4, we first establish the following lemma to characterize the estimation variance $\mathbb{E}_{\mathcal{B}}\left\|\frac{\partial\mathcal{L}_{\mathcal{D}}(\phi_k, \widetilde{w}_k^T; \mathcal{B})}{\partial\phi_k} - \frac{\partial\mathcal{L}_{\mathcal{D}}(\phi_k, \widetilde{w}_k^T)}{\partial\phi_k}\right\|^2$, where $\widetilde{w}_k^T$ is the output of $T$ inner-loop steps of gradient descent at the $k^{th}$ outer loop.

**Lemma 10.** *Suppose Assumptions 2 and 3 are satisfied and suppose each task loss $\mathcal{L}_{\mathcal{S}_i}(\phi, w_i)$ is $\mu$-strongly-convex w.r.t. $w_i$. Then, we have*

$$\mathbb{E}_{\mathcal{B}}\left\|\frac{\partial\mathcal{L}_{\mathcal{D}}(\phi_k, \widetilde{w}_k^T; \mathcal{B})}{\partial\phi_k} - \frac{\partial\mathcal{L}_{\mathcal{D}}(\phi_k, \widetilde{w}_k^T)}{\partial\phi_k}\right\|^2 \leq \left(1 + \frac{L}{\mu}\right)^2\frac{M^2}{|\mathcal{B}|}.$$

*Proof.* Let $\widetilde{w}_k^T = (w_{1,k}^T, ..., w_{m,k}^T)$ be the output of $T$ inner-loop steps of gradient descent at the $k^{th}$ outer loop. Using Proposition 2, we have, for task $\mathcal{T}_i$,

$$
\left\| \frac{\partial \mathcal{L}_{\mathcal{D}_i}(\phi_k, w_{i,k}^T)}{\partial \phi_k} \right\| \leq \| \nabla_\phi \mathcal{L}_{\mathcal{D}_i}(\phi_k, w_{i,k}^T) \|
$$

$$
+ \left\| \alpha \sum_{t=0}^{T-1} \nabla_\phi \nabla_{w_i} \mathcal{L}_{\mathcal{S}_i}(\phi_k, w_{i,k}^t) \prod_{j=t+1}^{T-1} (I - \alpha \nabla_{w_i}^2 \mathcal{L}_{\mathcal{S}_i}(\phi_k, w_{i,k}^j)) \nabla_{w_i} \mathcal{L}_{\mathcal{D}_i}(\phi_k, w_{i.k}^T) \right\|
$$

$$
\overset{(i)}{\leq} M + \alpha LM \sum_{t=0}^{T-1} (1 - \alpha\mu)^{T-t-1} = M + \frac{LM}{\mu}, \tag{72}
$$

where $(i)$ follows from assumptions 2 and strong-convexity of $\mathcal{L}_{\mathcal{S}_i}(\phi, \cdot)$. Then, using the definition of $\mathcal{L}_{\mathcal{D}}(\phi, \widetilde{w}; \mathcal{B}) = \frac{1}{|\mathcal{B}|} \sum_{i \in \mathcal{B}} \mathcal{L}_{\mathcal{D}_i}(\phi, w_i)$, we have

$$
\mathbb{E}_{\mathcal{B}} \left\| \frac{\partial \mathcal{L}_{\mathcal{D}}(\phi_k, \widetilde{w}_k^T; \mathcal{B})}{\partial \phi_k} - \frac{\partial \mathcal{L}_{\mathcal{D}}(\phi_k, \widetilde{w}_k^T)}{\partial \phi_k} \right\|^2 = \frac{1}{|\mathcal{B}|} \mathbb{E}_i \left\| \frac{\partial \mathcal{L}_{\mathcal{D}_i}(\phi_k, w_{i,k}^T)}{\partial \phi_k} - \frac{\partial \mathcal{L}_{\mathcal{D}}(\phi_k, \widetilde{w}_k^T)}{\partial \phi_k} \right\|^2
$$

$$
\overset{(i)}{\leq} \frac{1}{|\mathcal{B}|} \mathbb{E}_i \left\| \frac{\partial \mathcal{L}_{\mathcal{D}_i}(\phi_k, w_{i,k}^T)}{\partial \phi_k} \right\|^2
$$

$$
\overset{(ii)}{\leq} \left(1 + \frac{L}{\mu}\right)^2 \frac{M^2}{|\mathcal{B}|}. \tag{73}
$$

where $(i)$ follows from $\mathbb{E}_i \frac{\partial \mathcal{L}_{\mathcal{D}_i}(\phi_k, w_{i,k}^T)}{\partial \phi_k} = \frac{\partial \mathcal{L}_{\mathcal{D}}(\phi_k, \widetilde{w}_k^T)}{\partial \phi_k}$ and $(ii)$ follows from eq. (72). Then, the proof is complete. $\qquad\square$

**Proof of Theorem 4.** Recall $\Phi(\phi) := \mathcal{L}_{\mathcal{D}}(\phi, \widetilde{w}^*(\phi))$ be the objective function, and let $\widehat{\nabla}\Phi(\phi_k) = \frac{\partial \mathcal{L}_{\mathcal{D}}(\phi_k, \widetilde{w}_k^T)}{\partial \phi_k}$. Using an approach similar to eq. (42), we have

$$
\Phi(\phi_{k+1}) \leq \Phi(\phi_k) + \langle \nabla\Phi(\phi_k), \phi_{k+1} - \phi_k \rangle + \frac{L_\Phi}{2} \|\phi_{k+1} - \phi_k\|^2
$$

$$
\leq \Phi(\phi_k) - \beta \left\langle \nabla\Phi(\phi_k), \frac{\partial \mathcal{L}_{\mathcal{D}}(\phi_k, \widetilde{w}_k^T; \mathcal{B})}{\partial \phi_k} \right\rangle + \frac{\beta^2 L_\Phi}{2} \left\| \frac{\partial \mathcal{L}_{\mathcal{D}}(\phi_k, \widetilde{w}_k^T; \mathcal{B})}{\partial \phi_k} \right\|^2. \tag{74}
$$

Taking the expectation of eq. (74) yields

$$
\mathbb{E}\Phi(\phi_{k+1}) \overset{(i)}{\leq} \mathbb{E}\Phi(\phi_k) - \beta\mathbb{E}\langle \nabla\Phi(\phi_k), \widehat{\nabla}\Phi(\phi_k) \rangle + \frac{\beta^2 L_\Phi}{2} \mathbb{E}\|\widehat{\nabla}\Phi(\phi_k)\|^2
$$

$$
+ \frac{\beta^2 L_\Phi}{2} \mathbb{E} \left\| \widehat{\nabla}\Phi(\phi_k) - \frac{\partial \mathcal{L}_{\mathcal{D}}(\phi_k, \widetilde{w}_k^T; \mathcal{B})}{\partial \phi_k} \right\|^2
$$

$$
\overset{(ii)}{\leq} \mathbb{E}\Phi(\phi_k) - \beta\mathbb{E}\langle \nabla\Phi(\phi_k), \widehat{\nabla}\Phi(\phi_k) \rangle + \frac{\beta^2 L_\Phi}{2} \mathbb{E}\|\widehat{\nabla}\Phi(\phi_k)\|^2 + \frac{\beta^2 L_\Phi}{2} \left(1 + \frac{L}{\mu}\right)^2 \frac{M^2}{|\mathcal{B}|}
$$

$$
\leq \mathbb{E}\Phi(\phi_k) - \left(\frac{\beta}{2} - \beta^2 L_\Phi\right) \mathbb{E}\|\nabla\Phi(\phi_k)\|^2 + \left(\frac{\beta}{2} + \beta^2 L_\Phi\right) \mathbb{E}\|\nabla\Phi(\phi_k) - \widehat{\nabla}\Phi(\phi_k)\|^2
$$

$$
+ \frac{\beta^2 L_\Phi}{2} \left(1 + \frac{L}{\mu}\right)^2 \frac{M^2}{|\mathcal{B}|}, \tag{75}
$$

where $(i)$ follows from $\mathbb{E}_{\mathcal{B}} \mathcal{L}_{\mathcal{D}}(\phi_k, \widetilde{w}_k^T; \mathcal{B}) = \mathcal{L}_{\mathcal{D}}(\phi_k, \widetilde{w}_k^T)$ and $(ii)$ follows from Lemma 10. Using Lemma 6 in eq. (75) and rearranging the terms, we have

$$
\frac{1}{K} \sum_{k=0}^{K-1} \left(\frac{1}{2} - \beta L_\Phi\right) \mathbb{E}\|\nabla\Phi(\phi_k)\|^2
$$

$$
\leq \frac{\Phi(\phi_0) - \inf_\phi \Phi(\phi)}{\beta K} + 3\left(\frac{1}{2} + \beta L_\Phi\right) \frac{L^2 M^2 (1 - \alpha\mu)^{2T}}{\mu^2} + \frac{\beta L_\Phi}{2} \left(1 + \frac{L}{\mu}\right)^2 \frac{M^2}{|\mathcal{B}|}
$$

$$
+ 3\Delta\left(\frac{1}{2} + \beta L_\Phi\right) \left(\frac{L^2(L + \mu)^2}{\mu^2}(1 - \alpha\mu)^T + \frac{4M^2 (\tau\mu + L\rho)^2}{\mu^4}(1 - \alpha\mu)^{T-1}\right),
$$

where $\Delta = \max_k \|\widetilde{w}_k^0 - \widetilde{w}^*(\phi_k)\|^2 < \infty$. Choose the same parameters $\beta, T$ as in Theorem 2. Then, we have

$$\frac{1}{K} \sum_{k=0}^{K-1} \mathbb{E}\|\nabla\Phi(\phi_k)\|^2 \leq \frac{16L_\Phi(\Phi(\phi_0) - \inf_\phi \Phi(\phi))}{K} + \frac{2\epsilon}{3} + \left(1 + \frac{L}{\mu}\right)^2 \frac{M^2}{8|\mathcal{B}|}.$$

Then, the proof is complete. $\qquad\square$

