# OpenReview forum: "Provably Faster Algorithms for Bilevel Optimization and Applications to Meta-Learning"
_ICLR.cc/2021/Conference — Reject_

### Official Review · AnonReviewer4 · 2020-10-27
**The paper proposes two algorithms to solve bilevel optimization problems, with applications to hyperparameter optimization and meta-learning.**

**Rating:** 3
**Confidence:** 2

**Review:**

Overall, I vote for rejecting. While most of the related literature focuses on computing the hypergradient, this paper tackles the full optimization problem, which is much harder and more interesting. In particular I really like the idea of taking advantage of the (finite sum) structure of the bilevel optimization problem (algorithm 2). The problem tackled is very important but I have a lot of concerns regarding the novelty of the 'proposed' algorithm 1. Moreover I do not understand the focus on avoiding to compute the inverse of the Hessian, since in such application the Hessian is never inverted, only linear systems are solved, which is much cheaper.

##########################################################################

Advantages of the paper:
- The paper is overall well written.
- The idea of algorithm 2 is elegant.

##########################################################################

Concerns
1- Authors may have missed an important reference [1], which is a seminal paper on bilevel optimization for hyperparameter optimization.

2- Authors claim novelty on algorithm 1 'design a new bilevel optimization algorithm'.
I may have missed something important, but it seems to me that algorithm 1 not new: it is exactly the algorithm in [1], Section 4.

3- Moreover how do the authors chose the stepsize of the gradient descent $\beta$? IMO this is a strong bottelneck in practice, that is why [2] uses LBFGS once the hypergradient is computed, and [3] uses a line-search procedure.
Do authors have heuristics to choose the stepsizes $\beta$ in practice?
This is paramount in practice, and the value of the stepsize is hidden in appendix, set to '0.001'. This seems very custom, and not general. How do authors know that the proposed algorithms converged for this choice of stepsize?

4- 'proposition shows that the differentiation involves computations of second order derivatives'
I do not understand. To my knowledge, the hypergradient $\partial f$ can be computed in 3 differents ways: implicit[3], forward or backward [4].
To my knowledge there is not Hessian inversion, neither in the implicit nor in the backward: algorithm 1 in [3] solves a linear system to compute the gradient, and no Hessian are inverted in [1], Section 4.
I am very confused with this Hessian inversion problem. Since I did not understand it, I lost a lot of the motivation for the introduction of algorithm 2.

5- At the beginning of section 2.1 it seems that the end of an old sentence remains in the text.

6- Part 4 is very technical and felt to me like an arid succession of theoretical results. IMO more insights and comments on the theoretical results would be very helpful for the reader.

7- Part 5 proposes experiments using algorithm 1, which is already known (unless I missed something). Experiments on algorithm 2 are only in appendix.
A comparison with  the approximated procedure of [3] would strength the paper: from my experience, approximated gradient like in [3] drastically improves the convergence time.


[1]  Domke, J. Generic methods for optimization-based modeling, AISTAST 2012
[2] Deledalle, C. A., Vaiter, S., Fadili, J., & Peyré, G. (2014). Stein Unbiased GrAdient estimator of the Risk (SUGAR) for multiple parameter selection. SIAM Journal on Imaging Sciences
[3] Pedregosa, F. (2016). Hyperparameter optimization with approximate gradient, ICML2016
[4] Franceschi, L., Donini, M., Frasconi, P., & Pontil, M. (2017). Forward and reverse gradient-based hyperparameter optimization. ICML2017

---

> ### Author Response · Authors · 2020-11-21
> **Response to reviewer #4**
>
> Many thanks for the most helpful review! Based on the review comments, we have added a significant amount of new results both in theory and practice, via which we hope that we have addressed the reviewer’s previous concerns satisfactorily. We will appreciate it very much if the reviewer can go over these new results (as we explain below) and re-evaluate the paper. The updated PDF of our paper has been uploaded.
>
> $\bf Q1:$ Missed an important reference [1]. Some suggestions on other related works.
>
> $\bf A:$ Many thanks! We have added this reference and other mentioned related works in the revision (see Introduction and Section 1.2).
>
> $\bf Q2:$ Algorithm 1 is not new.
>
> $\bf A:$ Many thanks! We have modified Algorithm 1 as existing two types of bilevel optimizers based on approximate implicit gradient (AID) and iterative differentiation (ITD), which we call as AID-BiO and ITD-BiO, respectively. In the revision, we clarified this point and removed our previous claim of the design novelty of Algorithm 1 (see abstract, introduction and Section 2.1).
>
> Instead, in the revision, we managed to add new and comprehensive theoretical analysis (see Theorems 1 and 2, with new proofs in pages 17-20 and 21-23) for these two major existing approaches, which serves as our new contribution. In particular, our result improves the existing analysis for AID-BiO order-wisely via a warm start strategy as well as a more practical parameter selection, and our analysis for ITD-BiO is new in the literature.
>
> $\bf Q3:$ how do the authors choose the stepsize of the gradient descent $\beta$?
>
> $\bf A:$ Great question! Yes, the choice of $\beta$ is important. In our experiment, we choose it based on the following two procedures. First, we set up an initial value for $\beta$ based on the suggested hyperparameter setting of benchmark algorithms. For example, for our meta-learning experiments on few-shot learning (i.e., experiments in Figure 1), the existing repositories of ANIL and MAML set this $\beta$ to be 0.001, and hence we initialize $\beta$ to be $0.001$ for our method. Second, we finetune this value via a grid search (e.g., choose $\beta$ from {0.0001, 0.001,0.01,0.1}) on training and validation datasets to achieve a good performance and fast convergence. We have clarified it in Section A.2 and Section B (in the supplementary materials).
>
> We thank the reviewer for pointing out LBFGS and line-search methods for us, which can further improve the performance of our methods, but of course with some additional cost. We would like to incorporate such techniques in our method in the future work, which we believe deserves in-depth study both in theory and practice.
>
> $\bf Q4:$ About the Hessian inversion problem.
>
> $\bf A:$ Sorry about the confusion! Yes, there is no Hessian inversion done in the implementation, and we use the product of Hessian inverse $H^{-1}$  and the gradient $\nabla_yf(x,y)$ to denote the solution of the linear system $Hv = \nabla_yf(x,y)$ (in our setting, this Hessian is invertible and hence H^{-1} exists). In other words, we use Hessian inverse H^{-1} only for notation simplification. In our revision, v_Q in eq. (5) serves as an approximate solution for this linear system, and only Hessian-vector product rather than Hessian inversion needs to be computed (see Algorithm 3). As a result, our Algorithm 2 is computationally and memory efficient due to the data sampling.
>
> $\bf Q5:$ At the beginning of section 2.1 it seems that the end of an old sentence remains in the text.
>
> $\bf A:$ Many thanks! We have revised it.
>
> $\bf Q6:$ Part 4 is very technical and felt to me like an arid succession of theoretical results. IMO more insights and comments on the theoretical results would be very helpful for the reader.
>
> $\bf A:$ Many thanks! We have added comments and insights of the theoretical results in Section 4. For example, in contrast to existing analysis, our analysis motivates a choice of fixed number of inner-loop steps as well as a warm start strategy. We demonstrate such insight in our meta-learning experiments (see Figure 1). More details can be found in our updated version of this paper.
>
> $\bf Q7:$ Part 5 proposes experiments using algorithm 1, which is already known (unless I missed something). Experiments on algorithm 2 are only in appendix. A comparison with the approximated procedure of [3] would strengthen the paper: from my experience, approximated gradient like in [3] drastically improves the convergence time.
>
> $\bf A:$ For Algorithm 1, see our answer to the second question Q2.
>
> For Algorithm 2, many thanks for this great suggestion. We have added the optimizer HOAG proposed by [3] in all hyperparameter optimization experiments (see Figures 2, 3 and 4 in Appendix A.2 in supplementary materials) in the updated version of this paper. You can see that our proposed stocBiO is more efficient than HOAG, and such an improvement holds for a wide range of batch sizes.

---

### Official Review · AnonReviewer3 · 2020-10-28
**This is a paper that maybe cannot meet the acceptance criteria.**

**Rating:** 5
**Confidence:** 4

**Review:**

This paper proposes two novel algorithms, named deterBio and stocBio, for the nonconvex-strongly-convex bilevel optimization problems, and presents a comparison with several existing algorithms to demonstrate their superiority by experiments. Besides, propositions and theorems are well proved in the supplementary materials.

+ The idea of the proposal for reliable bilevel optimizers with better gradient complexities is meaningful.
+ The paper is well organized attached with sufficient supplementary materials for different propositions, and give analysis about the gradient complexities of proposed algorithms and related methods.
+ It gives a detailed description of the connection between conventional bilevel optimization and meta-learning problems and applications of respective fields.



Concerns:
- Duplicate ‘and’ appears in the last line of the 1st Page.
- The involved batch size and parameters are significant and sensitive to different problems, and the comparison between stocBio and other algorithms maybe not enough, perhaps more analyses are recommended.
- Seems that the experiments are not enough to prove the flexible applications in relative deep learning fields, such as reinforcement learning.

---

> ### Author Response · Authors · 2020-11-21
> **Response to reviewer #3**
>
> Many thanks for the most helpful review! Based on the review comments, we have added a significant amount of new results both in theory and practice, via which we hope that we have addressed the reviewer’s previous concerns satisfactorily. We will appreciate it very much if the reviewer can go over these new results (as we explain below) and re-evaluate the paper. The updated PDF of our paper has been uploaded.
>
> $\bf Q:$ Duplicate ‘and’ appears in the last line of the 1st page.
>
> $\bf A:$ Thanks for pointing this out! We have corrected it in the revision.
>
> $\bf Q:$ The involved batch size and parameters are significant and sensitive to different problems.
>
> $\bf A:$ Our experiments in Appendix A.3 show that the performance of our proposed stocBiO is not very sensitive to the batch size; stocBiO outperforms the competing algorithms under a wide range of batch sizes. Furthermore, other parameters in our algorithms are also easy to tune in our experiments on different problems. In specific, we adopt a simple two-stage procedure for parameter selection: 1) we set up initial values for all parameters based on the suggested hyperparameter setting of benchmark algorithms; 2) we use a grid search to finetune these parameters on training and validation data.
>
> $\bf Q:$ The comparison between stocBio and other algorithms may be not enough, perhaps more analyses are recommended.
>
> $\bf A:$ Thanks for this suggestion! For stochastic bilevel optimization, we have added a new experiment on logistic regression problem over 20 Newsgroup. In this experiment, we compare our stocBiO with an extensive list of benchmark algorithms including BSA, reverse, AID-FP, AID-CG, HOAG. The results show that our stocBiO outperforms these methods by a large margin. Such an improvement holds for a wide range of batch sizes.
> Furthermore, we have also updated the data hyper-cleaning experiment, and added a competitor HOAG [Pedregosa, 2016] in the comparison. The results still show that our proposed stocBiO performs the best among all algorithms at different corruption rates.

---

### Official Review · AnonReviewer1 · 2020-10-28
**Reviewer 1**

**Rating:** 6
**Confidence:** 2

**Review:**

The paper presents two algorithms - one for the deterministic and one for stochastic bilevel optimization. The paper claims the methods are lower cost in computational complexity for various terms and easy to implement. A finite-time convergence proof is provided for the algorithms.  Empirical results are presented for meta-learning, and (in the appendix) hyperparameter optimization.


Strengths:

The deterministic bilevel algorithm is easy to implement and has a convergence proof given standard assumptions.

The stochastic bilevel algorithm has a convergence proof given standard assumptions.

They investigate in-depth how the complexity of their algorithms scales with various terms (GC, Hxy, Hyy, etc) showing benefits on some.


Weaknesses:

The experiments presented in the paper are limited, and difficult to assess even after looking through the appendix.  What do the low-alpha parts of Figure 1 represent?  Why is CG used for the inverse with BA?  The algorithm you investigate (a Neumann Hessian inverse) was shown to work better in these kinds of problems by [1], [2], [3].

Algorithm 3 is not useful for practitioners to implement.  It seems like it does not use hessian-vector products, so I doubt the current incarnation would scale to any reasonably sized problem. See [1] or [2] for papers that provide memory-efficient Hessian Inverse estimators with the Neumann series and Hessian-vector products. Does Table 2 rely on storing the full Hessians in memory?  If so, this is infeasible in most applications and the complexity comparisons are not useful.

[1] Lorraine, Jonathan, Paul Vicol, and David Duvenaud. "Optimizing millions of hyperparameters by implicit differentiation." International Conference on Artificial Intelligence and Statistics. PMLR, 2020.
[2] Liao, Renjie, et al. "Reviving and improving recurrent back-propagation." arXiv preprint arXiv:1803.06396 (2018).
[3] Shaban, Amirreza, et al. "Truncated back-propagation for bilevel optimization." The 22nd International Conference on Artificial Intelligence and Statistics. 2019.


My recommendation is to reject this work with a 4. The paper presents convergence proofs for variants of two common algorithms in bi-level optimization. If a version of algorithm 3 that uses Hessian-vector products is presented with similar convergence results I would change my score to a 6.  I would also increase the score if (quite a bit) stronger experimental results are presented -- the meta-learning results seem mediocre and all hyperparameter results are similar and in the appendix.  I am very familiar with this area, however, I worry I may have misunderstood some notation about the implementation, so I will put my confidence as lower than otherwise until author feedback.


The following points did not affect my score, but may help improve the paper:

The level of formality in the writing seems different than other submissions to ICLR.  For example, saying “etc.” or “and/or” in the abstract.

I am not familiar with the term “order-level lower” complexity.

“Inlcude” to “include” and “trails” to “trials”

I think “hyperparameter” may be a more common spelling than “hyper-parameter.”
Also, “backpropagation” may be more common than “back-propagation”

Passing the paper through a grammar checker might help catch a few other simple grammar typos.

---

> ### Author Response · Authors · 2020-11-21
> **Response to reviewer #1**
>
> Many thanks for the most helpful review! Based on the review comments, we have added a significant amount of new results both in theory and practice, via which we hope that we have addressed the reviewer’s previous concerns satisfactorily. We will appreciate it very much if the reviewer can go over these new results (as we explain below) and re-evaluate the paper. The updated PDF of our paper has been uploaded.
>
> $\bf Q:$ What do the low-alpha parts of Figure 1 represent?
>
> $\bf A:$ In case that we did not understand the reviewer’s question correctly (as we answer below), please kindly elaborate it further and we will respond readily. We assume that by “low-alpha parts of Figure 1” the reviewer refers to the legend of Figure 1 for each curve. These acronyms of ITD-BiO, AID-BiO-constant, AID-BiO-increasing refer to the algorithms explained at the beginning of Section 5.2, and MAML and ANIL are two algorithms explained via the footnotes at the bottom of page 9.
>
> $\bf Q:$ Why is CG used for the inverse with BA?
>
> $\bf A:$ We now use CG for both algorithms for a fair comparison. After revising our paper, the goal of the meta-learning experiments is to validate our theoretical results that AID-BiO (which stands for bilevel optimizer via approximate implicit gradient (AID)) converges faster under the parameters suggested by our analysis than that in Ghadimi and Wang, 2018. As you can see in the new Figure 1, we show that AID-BiO-constant (our analysis) is faster than AID-BiO-increasing (i.e., BA in Ghadimi and Wang, 2018).
>
> $\bf Q:$ Algorithm 3 is not practical.
>
> $\bf A:$ Many thanks for this great suggestion! We have updated Algorithm 3 in the revision, which now involves only Jacobian- and Hessian-vector product computations, and hence are scalable and efficient in practice. We have updated and added experiments In Section A.2 and A.3 (in appendix), where you can see that the proposed stocBiO outperforms the existing benchmark bilevel optimizers including AID-CG, AID-FP, reverse, HOAG, BSA by a large margin. The codes for our implementations have been uploaded.
>
> We also provided the new convergence result for the improved algorithm (see Proposition 3 and Theorem 3 with their proofs in pages 23-25 and 25-30), which achieves better computational complexity guarantee than that in our initial submission.
>
> The complexity results, i.e., Tables 1 and 2, are measured in terms of the number of Jacobian- and Hessian-vector products, rather than the number of second-order derivatives, e.g, Hessians. Thus, these results are more practically meaningful.
>
> $\bf Q:$ Suggestions on the formality and typos in the writing.
>
> $\bf A:$ Many thanks for these helpful suggestions! We have proofread the paper carefully, and revised our formality and all typos accordingly in the writing.
>
> $\bf Q:$ About “order-level lower” complexity.
>
> $\bf A:$ The complexity of an algorithm captures how the number of computations (in terms of the total number of gradient, Jacobian- and Hessian-vector product computations) scales with the required accuracy $\epsilon$, where $\epsilon$ is typically very small. Here, the scaling is with respect to $1/\epsilon$, meaning higher accuracy (i.e., smaller $\epsilon$) requires more computations. “Order-level lower complexity” means that an algorithm requires an order of magnitude less amount of computations in terms of $1/\epsilon$ to achieve an $\epsilon$-accurate point, which represents a significant reduction of computations.

---

### Official Review · AnonReviewer2 · 2020-10-29
**The paper propose algorithm for solving bilevel opt. with gradient descent (computed with back-propagation). It provides a clear complexity analysis. However, its novelty wrt previously introduced hypergradient computation is a bit fuzzy to me.**

**Rating:** 7
**Confidence:** 3

**Review:**



The paper propose two algorithms for solving bilevel optimization problem where the inner objective is assumed to be strongly convex. The main idea is to use an iterative scheme to approximate the inner level solution, and apply to chain rule to compute the upper level gradient à la back-propagation. Similar techniques is used on a stochastic version in case where the objective function are written as an expectation. The authors also provide a nice complexity analysis (under relatively restrictive but common assumptions) and numerical experiments which confirm an improvement upon some others previous methods eg (Ghadimi and Wang, 2018). The paper is reasonably well written.

I have only one concern on the novelty:

- the paper (Franceschi etal, 2017, 2018) computes hypergradient by backpropagation (the reverse-mode) as well and the actual paper does not clearly describe the relation with these works (while claiming efficiency superiority). More there is no comparisons, neither theoretical nor practical. Note that its iteration complexity was recently provided in ICML (Grazzi etal, 2020). It should be interesting that the authors precisely comment on the differences and add this competitor in the numerical experiments. Specially clarification on the differences (if any) between hypergradient formula would be helpful, as well as the time and space complexity of the proposed algorithms.

I am willing to raise my score after precise clarification of this point.

- Also, it would be interesting to provide an easy to use open source implementation for the conference venue and futur comparisons.

Typos:
"and and" on page 1.
"fist" on page 2.

---

> ### Author Response · Authors · 2020-11-21
> **Response to reviewer #2:**
>
> Many thanks for the most helpful review! Based on the review comments, we have added a significant amount of new results both in theory and experiments, via which we hope that we have addressed the reviewer’s previous concerns satisfactorily. We will appreciate it very much if the reviewer can go over these new results (as we explain below) and re-evaluate the paper. The updated PDF of our paper has been uploaded.
>
> $\bf Q:$ one concern on the novelty about Algorithm 1.
>
> $\bf A:$ We thank the reviewer for pointing this out for us! Yes, Algorithm 1 in our initial submission uses the same iterative differentiation technique (i.e., the backpropagation approach) as in Domke, J, 2012, Franceschi et al., 2017, 2018, for hypergradient computation. In the revision (see abstract, introduction and Section 2.1), we clarified this point and removed our previous claim of the design novelty of Algorithm 1.
>
> Instead, in the revision, we managed to add new and comprehensive theoretical analysis (see Theorems 1 and 2, new proofs are in pages 17-20 and 21-23 respectively) for these two popular types of approaches based on iterative differentiation (ITD) and approximate implicit differentiation (AID), which we call as ITD-BiO and AID-BiO. As pointed out by the reviewer, Grazzi et al, 2020 analyzed only the iteration complexity of the hypergradient estimation for ITD-BiO and AID-BiO. As a comparison, we provided the finite-time convergence and complexity analysis for the entire execution of the ITD-BiO and AID-BiO algorithms, which requires to further deal with the double-loop structure, backpropagate tracking errors of hypergradient approximation to previous loops, and design/selection of various parameters (e.g., stepsizes, inner-loop length) to guarantee the overall convergence. We have also revised and refined our meta-learning experiments (see Figure 1) to validate the theoretical results, e.g., demonstrate the effectiveness of the theoretically suggested parameter selection.
>
> We also want to emphasize that our stocBiO (i.e., algorithm 2) for stochastic bilevel optimization contains novel designs and treatments, which do not exist for deterministic bilevel optimization. For example, one can easily see that $E_{\xi} [H(\xi)^{-1}]$ is not equal to $[ E_{\xi} H(\xi)]^{-1}$, and hence designing an unbiased hypergradient estimator is challenging for the stochastic setting. In this paper, we develop a sample-efficient and nearly-unbiased adaptive hypergradient estimator via an AID based approach (see equations (5), (6) and algorithm 3), which, importantly, involves only Jacobian- and Hessian-vector products rather than second-order derivatives.
>
> $\bf Q:$ it would be interesting to provide an easy to use open source implementation for the conference venue and future comparisons.
>
> $\bf A:$ Thanks for this suggestion! We have uploaded our codes for the implementation of our stocBiO algorithm and its comparison to other existing benchmark methods as a supplementary file. We will upload our cleaned codes to open source platforms such as GitHub in the near future.

---

> > ### Comment · AnonReviewer2 · 2020-11-25
> > **The authors clarified my main concern.**
> >
> > The authors did a great job of updating the paper and clarifying the new contributions (the analysis of the stochastic version is relevant and novel)
> > I will raise my score accordingly and vote for acceptance.

---

### Author Response · Authors · 2020-11-21
**Response to all reviewers:**

We thank all reviewers for their most valuable comments! Based on the review comments, we have added a significant amount of new results both in theory and experiments, via which we hope that we have addressed all of the reviewers’ previous concerns satisfactorily. We will appreciate it very much if the reviewers can go over these new results (as we explain below) and re-evaluate the paper. The revision has been uploaded, and all revised parts are marked by blue color. The proofs for all new theorems have been provided and the codes for the experiments have been uploaded. As a summary, we have added the following new results in the revision.

Thanks to the reviewers’ comments, we have revised Algorithm 1 as a unified algorithm, which can take either popular iterative differentiation (ITD) or implicit differentiation (AID) (i.e., the existing two types of techniques for hypergradient computation). In the revision, we remove our previous claim of the design novelty of Algorithm 1, which is now attributed to the previous studies.

(1):  $\text{\bf New Theorems 1 and 2 with new proofs in pages 17-20 and 21-23.}$ In our revision, we provided a new and comprehensive finite-time analysis in Theorems 1 and 2 for the aforementioned two types of approaches, which we believe serve as new and novel theoretical contributions to bilevel optimization. For the AID method, our theory improves the previous convergence analysis for AID with an order-level improvement (see Table 1), due to a more practical parameter selection as well as a novel warm start strategy. For the ITD method, to the best of our knowledge, our new theorem here for the first time establishes the finite-time convergence for this method. Furthermore, our analysis also provides a quantitative comparison between ITD and AID based methods (see Tables 1).

(2): $\text{\bf Improving Alg 2 (stocBiO) and Alg 3 (sub-alg of Alg 2).}$ For stochastic bilevel optimization, we provided an efficient and scalable version of Alg 3 (which is a sub-algorithm of Alg 2) by computing only Hessian-vector products. We also improve the overall algorithm stocBiO (i.e., Alg 2) by computing only Jacobian- and Hessian-vector products rather than second-order derivatives.

(3): $\text{\bf Updated Proposition 3 and Theorem 3 with proofs in pages 23-25 and 25-30.}$ For the improved stocBiO (i.e., Alg 2), we provide the new convergence analysis in Proposition 3 and Theorem 3, which achieve better computational complexity guarantee than that in our initial submission.

(4): $\text{\bf Revised and refined meta-learning experiments in Figure 1.}$ We verify that AID-BiO with our suggested parameter selection achieves a much better performance than AID-BiO in Ghadimi and Wang (2018) that requires to continuously increase the number of inner-loop steps to guarantee the performance.

(5): $\text{\bf New experiments and more extensive comparison for stocBiO in Figures 2 and 3.}$ We have added a new experiment on logistic regression problem over 20 Newsgroup to compare our stocBiO with several benchmark algorithms including BSA [Ghadimi & Wang, 2018], reverse [Franceschi et al., 2017], AID with fixed-point method (AID-FP) [Grazzi et al., 2020], AID with conjugate gradient (AID-CG) [ Grazzi et al., 2020], hyperparameter optimization algorithm with approximate gradient (HOAG) [Pedregosa, 2016]. The results in Figures 2 and 3 in Appendix A.2 show that our stocBiO outperforms these methods by a large margin, and such an improvement holds for a wide range of batch sizes.

(6): $\text{\bf New experimental comparison with one more benchmark in Figure 4.}$ We have updated the data hyper-cleaning experiment with an additional competitor HOAG [Pedregosa, 2016] in the comparison. Figure 4 in Appendix A.2 shows that our proposed stocBiO performs the best among all algorithms at different corruption rates.

(7): $\text{\bf Our codes for stocBiO and the above comparisons have been uploaded.}$

---

### Decision · Program_Chairs · 2021-01-07
**Final Decision**

**Decision:**

Reject

**Comment:**

The authors propose two algorithms and their theoretical analysis for solving bilevel optimization problems where the inner objective is assumed to be strongly convex. The authors have greatly improved the paper to answer reviewer comments and three out of four reviewers have increased their scores. That said, given the large amount of new material added to this paper during the discussion phase, the program committee believes the paper requires a new round of reviews for a confident assessment. We encourage the authors to resubmit their work to a top conference such as ICML.